

# Changes in productivity and intermediate circulation in the northern Indian Ocean since the last deglaciation: new insights from benthic foraminiferal Cd/Ca records and benthic assemblage analyses

Ruifang Ma[1,2*], Sophie Sépulcre[1], Laetitia Licari[3], Frédéric Haurine[1], Franck Bassinot[4], Zhaojie Yu[5], Christophe Colin[1]

[1] GEOPS, Université Paris-Saclay, CNRS, Orsay, 91405, France.
[2] State Key Laboratory of Cryospheric Science, Northwest Institute of Eco-Environment and Resources, Chinese Academy of Sciences, Lanzhou, 730000, China.
[3] CEREGE, Aix-Marseille Université-Europole de l'Arbois-BP80, Aix-en-Provence, 13545, France.
[4] LSCE/IPSL, CEA CNRS UVSQ, Gif Sur Yvette, F-91190, France.
[5] Key Laboratory of Marine Geology and Environment, Institute of Oceanology, Chinese Academy of Sciences, Qingdao, 266071, China.

*Correspondence to: R. MA (maruifang89@hotmail.com)

**Abstract.** We have measured Cd/Ca ratios of several benthic foraminiferal species and studied benthic foraminiferal assemblages on two cores from the northern Indian Ocean (Arabian Sea and northern Bay of Bengal, BoB), in order to reconstruct variations in intermediate water circulation and paleo-nutrient content since the last deglaciation. Intermediate water $Cd_w$ records estimated from the benthic Cd/Ca reflect past changes in surface productivity and/or intermediate-bottom water ventilation. The benthic foraminiferal assemblages are consistent with the geochemical data. These results suggest that during the last deglaciation, the Heinrich Stadial 1 and Younger Dryas (HS1 and YD, respectively) millennial-scale events were marked by a decrease in $Cd_w$ values, indicating an enhanced ventilation of intermediate-bottom water masses. Benthic foraminifer assemblages indicate that surface primary productivity was low during the early Holocene (from 10 to 6 cal kyr BP), resulting in low intermediate water $Cd_w$ at both sites. From ~ 5.2 to 2.4 cal kyr BP, the benthic foraminiferal assemblages indicate meso- to eutrophic intermediate water conditions, which correspond to high surface productivity. This is consistent with a significant increase in the intermediate water $Cd_w$ in the southeastern Arabian Sea and the northeastern BoB. The comparison of intermediate water $Cd_w$ records with previous reconstructions of past Indian monsoon evolution during the Holocene suggests a direct control of intermediate water $Cd_w$ by monsoon-induced changes in upper water stratification and surface primary productivity.

## 1. Introduction

During the last deglaciation, a two-step rapid increase in atmospheric $CO_2$ occurred during the 18-14.7 and 12.8-11.7 cal kyr BP time intervals (e.g., Monnin et al., 2001). Several studies suggest that variations in the Southern Ocean circulation contributed to these increases in atmospheric $CO_2$ by transferring deep ocean carbon to the upper ocean and atmosphere, through enhanced upwelling and increased northward penetration of the Antarctic Intermediate Water (AAIW) in all ocean basins (e.g., Marchitto et al., 2007; Anderson et al., 2009; Skinner et al., 2014). Different proxies have been used to reconstruct past changes in intermediate circulation,




such as radiocarbon activity ($\Delta^{14}C$) (e.g., Marchitto et al., 2007; Bryan et al., 2010), benthic $\delta^{13}C$ (e.g., Pahnke
and Zahn, 2005; Jung et al., 2009; Ma et al., 2019), foraminiferal $\varepsilon_{Nd}$ (e.g., Pahnke et al., 2008; Xie et al., 2012;
Yu et al., 2018) and benthic foraminifera Sr/Ca (Ma et al., 2020). These studies have focused on the close
relationship between enhanced ventilation in the Southern Ocean and rising atmospheric $CO_2$ during the last
deglaciation period. Furthermore, it has been shown that glacial-interglacial transfer of $CO_2$ between the oceans
and the atmosphere could also be linked to changes in the efficiency of the oceanic biological pump (Pichevin et
al., 2009; Ziegler et al., 2013; Bauska et al., 2016; Hertzberg et al., 2016; Jaccard et al., 2016; Yu et al., 2019),
which may contribute up to half of the observed $CO_2$ flux (Kohfeld, 2005).
The oceanic biological pump and nutrient upwelling are at least partly controlled by intermediate-deep water
circulation, contributing to the observed $CO_2$ changes (e.g., Toggweiler, 1999; Marchitto and Broecker, 2006).
To track past changes in the nutrient concentration of intermediate water masses, benthic foraminifera Cd/Ca has
been used in many recent studies (e.g. Came et al., 2008; Poggemann et al., 2017; Valley et al., 2017; Umling et
al., 2018); indeed, the benthic foraminifera Cd/Ca is a robust proxy of seawater cadmium concentrations ($Cd_w$)
(Boyle, 1988; 1992), which shows a positive linear correlation with labile nutrients (phosphate and nitrate) in the
modern ocean (e.g., Boyle et al., 1976; Boyle, 1988; Elderfield and Rickaby, 2000). The benthic foraminifera
incorporate Cd as a function of $Cd_w$ with a species-dependent partition coefficient (e.g., Tachikawa and
Elderfield, 2002). Thus, the Cd measured in the fossil tests reflects the paleo-nutrient concentrations of the
surrounding water masses, and can be used to investigate past changes in intermediate-to-deep ocean properties
(e.g., Boyle and Keigwin, 1982; Oppo and Fairbanks, 1987; Came et al., 2008; Poggemann et al., 2017; Valley et
al., 2017; Umling et al., 2018).
Complementary to the geochemical proxies, the type of benthic foraminifers and their abundance, both of
which are related to organic flux and ecosystem oxygenation, make benthic foraminifer assemblages a powerful
proxy for estimating past variations in bottom water conditions (e.g., Corliss et al., 1986; Schmiedl et al., 1998;
Almogi-Labin et al., 2000) in conjunction with organic matter fluxes to the seafloor (e.g., Altenbach et al., 1999;
Van der Zwaan et al., 1999; Fontanier et al., 2002; Caulle et al., 2015). Benthic foraminifera have been
successfully used as indicators of surface productivity, especially in high carbon flux regions (Schnitker, 1994).
By comparing past benthic foraminiferal assemblages to modern ones, changes in food supply and oxygen
concentrations of the bottom water can be reconstructed (e.g., Corliss, 1979; Peterson, 1984; Murgese and De
Deckker, 2005). Recently, the combining of benthic foraminiferal assemblages and geochemical proxies has
received increasing attention and have been used to reconstruct the evolution of surface productivity and
upwelling intensity in the Indian Ocean (e.g., Hermelin 1991, 1992; Hermelin and Shimmield, 1995; Den Dulk
et al., 1998; Murgese and De Deckker, 2005).
The Arabian Sea is one of the most productive regions of the ocean today (Banse, 1987; Marra and Barber,
2005). Surface productivity is dominated by the monsoon system, which has a strong impact on the distribution
and dynamics of stratification and vertical mixing (Lévy et al., 2007). Numerous studies have focused on the
reconstruction of the paleo-productivity of the Arabian Sea in relation to past changes in monsoon intensity (e.g.,
Prell and Kutzbach, 1987; Naidu and Malmgren, 1996; Gupta et al., 2003; Singh et al., 2006; 2011; Bassinot et
al., 2011; Saraswat et al., 2014). By contrast, little is known about the paleoproductivity of the BoB (Phillips et
al., 2014; Zhou et al., 2020). Furthermore, the evolution of the nutrient content of intermediate water masses
since the last deglaciation has never been reconstructed in the Indian Ocean, where only two low-resolution





Cd/Ca records are available for deep-water depths (Boyle et al., 1995), and, to our knowledge, none are available
for intermediate water depths.
In this study, we provide, for the first time, two benthic foraminifera Cd/Ca records at intermediate water
depths in the northern Indian Ocean (Arabian Sea and northern Bay of Bengal). These data make it possible to
estimate past changes in the nutrient content, since the last deglaciation, over the last 17 kyr BP. We have also
investigated benthic foraminiferal assemblages obtained from core MD77-191 (southeastern Arabian Sea) to
help us reconstruct the conditions at the seafloor. Combined with planktonic foraminiferal $\delta^{18}O$, benthic $\delta^{13}C$,
and Cd/Ca records obtained from the same core, as well as with results already published in the Bay of Bengal
(Ma et al., 2019; 2020), this study aims to document past variations in intermediate- and deep-water conditions
and to decipher their links with surface paleo-productivity and intermediate water ventilation.
**2. Material and modern hydrological setting**
We analyzed sediment core MD77-191 (07°30'N-76°43'E, 1254m) located in the Arabian Sea (off the
southern tip of India), and core MD77-176 (14°30'5N-93°07'6E, 1375m) retrieved in the northeastern Bay of
Bengal (BoB). These cores were collected in 1977 during the OSIRIS III cruise of the French N/O Marion
Dufresne (Fig. 1).
The age model of core MD77-191 was established by using accelerator mass spectrometry (AMS) $^{14}C$ dates
obtained on 9 monospecific samples of planktonic foraminifera *Globigerinoides bulloides* (Bassinot et al., 2011),
one sample of pteropods (Mléneck, 1997), and three samples of planktonic foraminifera *Globigerinoides ruber*
(Ma et al., 2020). The average sedimentation rate of core MD77-191 is about 53 cm.kyr$^{-1}$ and up to 90 cm.kyr$^{-1}$
during the Holocene, providing a high-resolution, continuous record since 17 cal kyr BP.
The age model of core MD77-176 was previously established by using 31 planktonic foraminifer (*G. ruber*)
AMS $^{14}C$ dates combined with the core MD77-176 oxygen isotope record obtained on planktonic foraminifera *G.*
*ruber*, which were correlated to the GISP2 Greenland ice core record (Marzin et al., 2013). Core MD77-176
displays high accumulation rates (average ~25 cm.kyr$^{-1}$ and up to 40 cm.kyr$^{-1}$ during the Holocene).
In the modern ocean, the surface waters of the Arabian Sea and BoB are characterized by seasonally reversing
currents that are driven by the monsoon winds (Fig1.a). The surface water masses shallower than 150 m in the
Arabian Sea are mainly Arabian Sea high Salinity Water (ASHS, 36.5 psu) (Talley et al., 2011). In the BoB, the
surface waters above 100 m are designated Bay of Bengal surface waters (BoBSW), which have a low salinity
(31 psu) due to large river inputs (Talley et al., 2011). Today, the northward extension of AAIW in the Indian
Ocean rarely reaches beyond 10°S (Lynch-Stieglitz et al., 1994). The sites of cores MD77-191 and MD77-176
are mainly bathed, therefore, by the North Indian Intermediate Water (Olson et al., 1993; Reid, 2003) with a
potential contribution from the Red Sea Outflow Water (RSOW) for the site MD77-191 (Beal et al., 2000) and
from the southern return flow of deep water (North Indian Deep Water) for the site of core MD77-176 (Talley et
al., 2011).
As far as surface waters are concerned, during the summer monsoon, the clockwise circulation in the
Arabian Sea drives high salinity waters from the northern to the southeastern Arabian Sea (Fig. 1 and S1). By
contrast, during the winter monsoon, the northeastern winds bring low salinity water (BoBSW) from the BoB.
The northern Indian Ocean, especially the Arabian Sea, is characterized by highly variable seasonal productivity





(Shankar et al., 2002). Southwest winds during the summer season induce a strong Ekman pumping resulting in
very active upwelling along the western coasts of the Arabian Sea and thus promoting strong surface
productivity (Shankar et al., 2002; Fig. S1). By contrast, the surface productivity in the BoB is generally weak
compared with the Arabian Sea (e.g., Prasanna Kumar et al., 2001; Thushara and Vinayachandran, 2016;
O'Malley, 2017; Fig. S1). In the BoB, large river inputs of fresh water and direct monsoon precipitation lead to
more stable stratification in the upper ocean (Vinayachandran et al., 2002), and hence the vertical mixing of
nutrients from the subsurface to the euphotic zone is generally limited (Gomes et al., 2000). However, the
primary productivity of the western BoB shows a slight increase during the winter monsoon, as indicated by the
distribution of chlorophyll in the surface water (Thushara and Vinayachandran, 2016; O'Malley, 2017; Fig. S1).
Modern data indicate that the southern-sourced intermediate water (AAIW) in the Indian Ocean has a
phosphate concentration of about 2-2.5 µmol/kg (Figs. 1b and c). In the Northern intermediate Indian Ocean, the
phosphate concentration is significantly higher, ranging from 2.75 to 3 µmol/kg in the Arabian Sea during the
summer monsoon, and from 2.5 to 2.75 µmol/kg in the BoB during the winter monsoon (Figs. 1b and c). The
higher phosphate in the northern Indian Ocean can been linked to primary productivity (Banse, 1987; Marra and
Barber, 2005).

**3. Methods**
**3.1. Cd/Ca analysis**

We analyzed Cd/Ca in three calcite (*Cibicidoides pachyderma*, *Uvigerina peregrina*, and *Globobulimina* spp.)
and one aragonite (*Hoeglundina elegans*) benthic foraminiferal species from core MD77-191. *C. pachyderma* is
an epifaunal species, *U. peregrina* and *Globobulimina* spp. are endobenthic species with intermediate and deep
microhabitats, respectively (Fontanier et al., 2002). In core MD77-176, due to the limitation of calcitic species,
we only measured Cd/Ca ratios in *H. elegans* shells.
Each sample contained between 10 and 15 individuals picked from the 250-315µm size fraction. Samples
were gently crushed, cleaned to remove clays, organic matter and elemental oxides by using reductive and
oxidative cleaning following previously published methods (Boyle and Keigwin, 1982; Barker et al., 2003).
Each sample was dissolved in 0.075N HNO$_3$ and analyzed using a single collector sector field high resolution
inductively coupled plasma mass spectrometer (HR-ICP-MS) Thermo Element XR hosted at the GEOPS
Laboratory (University Paris-Saclay, France).
The detailed instrumental settings and mother standard solutions are described in Ma et al., (2020). A blank
consisting of the same 0.1N HNO$_3$ used to dilute the standards and samples was also analyzed. We removed the
blank intensity values from all the raw intensities (including standards), and raw data were linearly drift-
corrected by interspersing a drift standard every four samples. Standard curves were used to calculate elemental
concentrations, coefficients of determination ($r^2$) always being >0.9999 for all elemental ratios. The mean
reproducibility and accuracy are 3.6% and 7.5%, respectively.

**3.2 Faunal analysis**

Benthic foraminiferal assemblages from core MD77-176 have already been published in Ma et al. (2019). For
core MD77-191, a total of 72 samples were collected for benthic foraminiferal assemblage determinations. In
each sample, benthic foraminifera (>150 µm) were extracted, counted and identified to species level following



the taxonomical descriptions of various authors (e.g., Loeblich and Tappan, 1988; Jones, 1994; Holbourn et al.,
2013). For core MD77-191, there is no material left in this old, low diameter core and so we used samples
obtained earlier for stable isotope studies. Since the bulk weights of these samples were not recorded prior to
sieving, we could not perform the calculation of absolute abundance of foraminifera or accumulation rates. Thus,
we only converted the individual counts to percentages with respect to the total benthic foraminifera present in
each sample. In order to describe major faunal variations, we performed principal component analysis (PCA)
using the PAST software (Version 3.0, Hammer et al., 2001). Species present with a percentage >1% in at least 1
sample were used for statistical analysis and diversity calculation.

## 4. Results

### 4.1. Elemental ratios results

To check the influence of oxide contaminants on the elemental ratios, Mn/Ca was systematically measured.
The Mn/Ca of *H. elegans* from cores MD77-191 and MD77-176 ranges between 6.5-10 µmol/mol and 1-30
µmol/mol, respectively. Such ranges are much lower than the 100 µmol/mol limit proposed by Boyle (1983).
The Mn/Ca obtained on the three calcite benthic foraminifera species from core MD77-191 - *C. pachyderma* (5-
18 µmol/mol), *U. peregrina* (3-23 µmol/mol) and *Globobulimina* spp. (4-69 µmol/mol) - are also all below 100
µmol/mol (Boyle, 1983). The Fe/Ca ratios are also lower than 1 mmol/mol in all samples from cores MD77-191
and MD77-176, in agreement with the limit proposed by Barker et al. (2003). In addition, Barker et al. (2003)
concluded that no significant pollution by clay minerals would be expected when Al/Ca is <0.5 mmol/mol. In all
our samples, Al/Ca is below 0.5 mmol/mol, indicating that the sample cleaning procedure was efficient.
All of the above results indicate that our samples were not affected by contamination.

### 4.1.1 Cd/Ca

The Cd/Ca records of *C. pachyderma*, *U. peregrina* and *Globobulimina* spp. from core MD77-191 range
between 0.07-0.2 µmol/mol, 0.07-0.14 µmol/mol and 0.03-0.09 µmol/mol, respectively (Figs. 2e-g;
supplementary Table S1).
The Cd/Ca records for the calcite benthic species *C. pachyderma* and *U. peregrina* have very low time
resolutions during the last deglaciation. However, some common patterns can be observed. The Cd/Ca records of
*C. pachyderma* and *U. peregrina* show lower values during the Heinrich stadial 1 (HS1, 17-15.2 cal kyr BP) and
Younger Dryas (YD, 12-11 cal kyr BP) cold periods, with average values of ~0.08 µmol/mol for *C. pachyderma*
and ~0.09 µmol/mol for *U. peregrina*. By contrast, these two species display higher Cd/Ca ratios (~0.12
µmol/mol) during the Bølling-Allerød warm period (B-A, 15-13.3 cal kyr BP) compared with the HS1 and YD.
Then, lower values (~0.1 µmol/mol for *C. pachyderma*; 0.11 µmol/mol for *U. peregrina*) are observed during the
early Holocene (10-5 cal kyr BP) compared to larger variations occurring in the late Holocene (5.2-2.4 cal kyr
BP). The Cd/Ca record of deep infaunal *Globobulimina* spp., obtained at a lower time resolution, shows different
variations compared with the two other taxa without any clear trend during the Holocene.
The *H. elegans* Cd/Ca values of core MD77-191 range from 0.05 to 0.31µmol/mol since 17 cal kyr BP (Fig.
2d; supplementary Table S1). Depleted values at about 0.07 µmol/mol are recorded from the last deglaciation to
the early Holocene (17-5 cal kyr BP time interval). During the HS1 and the YD time intervals over the last
deglaciation, significant decrease of about ~0.05 µmol/mol occurred (even when taking into consideration the



analytical error bar of ±0.02, 2σ), and a slight increase (0.09 μmol/mol) is observed between 15 and 13.3 cal kyr
BP (B-A period). A rapid increase in the Cd/Ca values beginning at 5.2 cal kyr BP reaches a maximum (0.31
μmol/mol) during the late Holocene.
For core MD77-176, the *H. elegans* Cd/Ca records range between 0.06 and 0.17 μmol/mol over the past 18 cal
kyr BP (Fig. 2h; supplementary Table S1), without no clear trends and average benthic Cd/Ca values of ∼0.09
μmol/mol during the different periods (HS1, YD and Holocene). However, the benthic Cd/Ca record during the
Holocene seems to exhibit a slight increase both in value and range of variations after 6 cal kyr BP.

**4.2. Foraminifera assemblages of core MD77-191**

Benthic foraminiferal species richness ranges between 16 and 36, and the total abundance fluctuates between
82 and 642 specimens (supplementary Table S2). Hyaline species are the dominant constituents (>80%), and
mainly consist of *Bulimina aculeata*, *H. elegans*, *C. pachyderma*, *Uvigerina* spp., *Gyroidina broeckhiana*,
*Globocassidulina subglobosa*, *Sphaeroidina bulloides*, *Gyroidinoides* spp., *Lenticulina* spp., *Melonis*
*barleeanum*, and *Globobulimina* spp. (including *Praeglobobulimina* spp.) (in decreasing order of relative
average abundance). Agglutinated taxa reach on average about 1.6%, and consist of *Textularia* sp.,
*Martinottiella communis,* and *Eggerella bradyi*. The average percentage of porcelaneous species, characterized
by *Pyrgo elongata*, *Pyrgo murrhina*, *Pyrgo depressa*, *Pyrgoella irregularis*, *Quinqueloculina* spp., *Sigmoilopsis*
*schlumbergeri*, and *Spiroloculina* spp., is about 5.1%.
Furthermore, we merged species that share an ecological similarity, such as *Globobulimina affinis*,
*Globobulimina pacifica*, and *Praeglobobulimina* spp. into *Globobulimina* spp. A total of 74 samples and 55
groups/species were adopted to perform principal component analysis (PCA) in order to identify major faunal
trends. The PCA analysis suggests that the benthic foraminifera could be grouped into three assemblages, and
represent about 61% of the total variance (Table 1).
The combination of *Bulimina aculeata* and *C. pachyderma*, together with *Pullenia bulloides* and
*Ehrenbergina trigona* (Figs. 3 and S2), display high positive PC1 loadings. This assemblage, referred hereafter
as assemblage 1, dominated the foraminiferal record during the late Holocene (between 6 and 1.4 cal kyr BP).
By contrast, *H. elegans* and *Bulimina manginata* exhibit high negative PC1 loadings, and dominate assemblage 2,
which corresponds to the record during the early Holocene (Figs. 3 and S2). Other quantitatively important
contributors are *C. wuellerstorfi* and *Globocassidulina subglobosa* (Fig. S2).
The total variance of PC2 is 19%; for the positive loadings of PC2, *Sphaeroidina bulloides* and *Gyroidinoides*
*orbicularis* dominate assemblage 3, which is more important during the last deglaciation (Figs. 3 and S2). The
main associated species of assemblage 3 are *Bulimina mexicana* and *Gyroidinoides soldanii* (Fig. S2). However,
the main species from negative loadings consist of *Bulimina aculeata*, *H. elegans* and *C. pachyderma*, which
dominated the Holocene. The main composition of PC2 negative loadings is dominated by the same benthic
species as assemblages 1 and 2, which, as we have seen above, correspond to the Holocene; it is difficult,
therefore, to glean any additional information from this regarding bottom conditions. For this reason we only
recognize three assemblages in this paper.



## 5. Discussion

### 5.1. Past intermediate water $Cd_w$ concentrations from the Northern Indian Ocean

In the modern ocean, benthic foraminifera Cd/Ca shows a positive correlation with $Cd_w$ and dissolved nutrients (phosphate and nitrate) (Boyle et al., 1976; Hester and Boyle, 1982). As aragonite benthic foraminifera *H. elegans* faithfully records the bottom water Cd concentrations ($Cd_w$), Cd/Ca ratios can be converted to seawater $Cd_w$ with the appropriate relationship (Eq.1), where the partition coefficient $D_p \approx 1$ for all water depths (Boyle et al., 1995; Bryan and Marchitto, 2010).

$$Dp = \frac{(Cd/Ca)_{foram}}{(Cd/Ca)_{water}} \quad (Eq.1)$$

In contrast, the partition coefficient for calcite species changes with water depth. For water depths between 1150-3000 m, $D_p$ was calculated based on the equation of Boyle, (1992; Eq. 2). The seawater Ca concentration is assumed to be at a constant, mean value of 0.01 mol/kg (Boyle, 1992).

$$D_p = 1.3 + (depth - 1150) \times (1.6/1850) \quad (Eq.2)$$

The intermediate $Cd_w$ results based on the *H. elegans* Cd/Ca values of core MD77-191, range from 0.5 to 3.1 nmol/kg since 17 cal kyr BP (Fig. 4a), with a core top value of 0.80 nmol/kg in agreement with the estimated intermediate water depth modern $Cd_w$ (~0.83 nmol/kg) in the northern Indian Ocean (Boyle et al., 1995). Variations of *H. elegans* $Cd_w$ during the last deglaciation indicate a decrease of about ~0.5 nmol/kg in the HS1 and YD periods, with a slight increase (0.9 nmol/kg) during the warm B-A. $Cd_w$ results from core MD77-191 indicate a shift from the last deglaciation (~0.8 nmol/kg) to the late Holocene (~1.59 nmol/kg). During the Holocene, the $Cd_w$ records display relatively low values of around 0.9 nmol/kg in the 10-6 cal kyr BP time interval, and show a major shift at around 6.4 cal kyr BP with values rising up to 3.1 nmol/kg.

The intermediate $Cd_w$ was also calculated from calcite benthic species *C. pachyderma*, *U. peregrina* and *Globobulimina* spp. from core MD77-191, with values ranging between 0.53-1.48 μmol/mol, 0.52-1.04 μmol/mol and 0.26-0.65 μmol/mol, respectively (Figs. 4b-d). The $Cd_w$ values of *C. pachyderma* and *U. peregrina* are within the same range. However, the deep infaunal *Globobulimina* spp. $Cd_w$ displays relatively much lower values and does not exhibit strong variations compared to the other species investigated in this study, displaying a general increasing trend from the last deglaciation to the Holocene. As *Globobulimina* spp. correspond to deep benthic infaunal species, this result may indicate a stable nutrient content of pore water, as compared to other benthic taxa associated with bottom water (Fig. 4d).

During the last deglaciation, the $Cd_w$ records of *C. pachyderma* and *U. peregrina* show a decreasing trend during the HS1 and YD events, with mean values of ~ 0.59 and 0.65 nmol/kg for *C. pachyderma* and ~ 0.62 and 0.67 nmol/kg for *U. peregrina*, respectively (Fig. 4b and c). The $Cd_w$ records all display higher values during the B-A, with average values of ~ 0.94 and 0.84 nmol/kg, respectively (Fig. 4b and c). The $Cd_w$ records show depleted values in the early Holocene, followed by an abrupt increase during the middle Holocene, with average values of ~0.87 nmol/mol for *C. pachyderma* and ~0.81 nmol/kg for *U. peregrina*.

Relative variations in the $Cd_w$ obtained from *C. pachyderma* and *U. peregrina* are in good agreement with the records obtained on *H. elegans*. However, the *H. elegans* $Cd_w$ values are higher than those from the two calcite



species, especially during the Late Holocene. Moreover, the core top data of *C. pachyderma* and *U. peregrina*
are also lower (~ 0.70 and 0.69 nmol/kg, respectively) than the modern estimated $Cd_w$ data (~ 0.83 nmol/kg) in
the northern Indian Ocean (Boyle et al., 1995). These depleted $Cd_w$ values may be related to the benthic
foraminiferal microhabitat effect; indeed, *U. peregrina* is known to be strictly a shallow infaunal species, as well
as *C. pachyderma* (Fontanier et al., 2002), differing from strictly epifaunal taxa, such as *Cibicides wuellerstorfi*
(Mackensen et al., 1993). Thus, when tracking past changes in the bottom water $Cd_w$ concentrations, the use of a
strictly epifaunal species living at the water-sediment interface such as *H. elegans* appears to be more robust than
using endofaunal species that live in contact with pore water.
For core MD77-176, the intermediate water $Cd_w$ calculated from the *H. elegans* Cd/Ca records ranges between
0.6 and 1.7 nmol/kg over the past 18 cal kyr BP (Fig. 4e). Compared with intermediate $Cd_w$ from MD77-191, the
$Cd_w$ record of core MD77-176 does not display any clear trend from the last deglaciation to the Holocene.
However, a slight increase is observed since 6 cal kyr BP, in agreement with the MD77-191 intermediate $Cd_w$
records. In addition, even though the MD77-176 record has a lower time resolution, it displays a shorter
maximum (1.3 nmol/kg) during the 13.4-11 cal kyr BP time interval.
To summarize, among the three calcite benthic taxa and the aragonitic benthic species *H. elegans*, the Cd/Ca
records of *H. elegans* appear to be the most suitable for tracking past $Cd_w$ changes at intermediate water depth
through time. Thus, in the following discussion, we will only focus on the intermediate $Cd_w$ calculated from the
*H. elegans* Cd/Ca from both studied cores.

**5.2. Comparison between geochemical records and benthic foraminiferal assemblages**

Comparing the geochemical records to the benthic assemblages, we can observe similar patterns. For core
MD77-191 from the southeastern Arabian Sea, three benthic assemblages were identified since the last
deglaciation. *S. bulloides* and *Gyroidinoides orbicularis* are major components of assemblage 3 (during the last
deglaciation), together with *B. mexicana* and *Gyroidinoides soldanii* (Figs. 3 and S2). *S. bulloides* and *B.*
*mexicana* are found in intermediate to high organic carbon flux rate regions (e.g., Schmiedl et al., 2000;
Eberwein and Mackensen, 2006, 2008), while *G. orbicularis* and *G. soldanii* are associated with well-
oxygenated and oligotrophic environments (Peterson, 1984; Burmistrova and Belyaeva, 2006; De and Gupta,
2010). Thus, assemblage 3 reflects mesotrophic environments and/or well-ventilated conditions during the last
deglaciation. Although millennial-scale changes in the benthic foraminiferal assemblages during the last
deglaciation could not be observed, benthic fauna 3 seems at least partly consistent with previous studies in the
northern Indian Ocean based on multiple geochemical proxies (e.g., benthic $\delta^{13}C$, intermediate water $[CO_3^{2-}]$ and
$\varepsilon_{Nd}$ records); these studies have revealed the presence of better-ventilated waters, which might correspond to
AAIW, during the HS1 and YD (e.g., Yu et al., 2018; Ma et al., 2019; 2020).
Benthic foraminiferal assemblage 2 predominates during the early Holocene and is characterized by *H.*
*elegans* and *B. manginata* as major contributors (Figs. 3 and S2). The other important contributors are *C.*
*wuellerstorfi* and *G. subglobosa. B. manginata* is found in high organic carbon flux rate conditions (De Rijk et
al., 2000; Eberwein and Mackensen, 2006, 2008). However, previous studies on *H. elegans*, *C. wuellerstorfi* and
*G. subglobsa* indicate that these species correspond to high levels of dissolved oxygen and oligotrophic settings
(e.g., Altenbach et al., 1999; Fontanier et al., 2002; Murgese and De Deckker, 2005, 2007; De and Gupta, 2010).
Periods dominated by these taxa probably indicate high oxygen levels and an oligotrophic environment.





Additionally, glacial to Holocene benthic $\delta^{13}C$ shifts (0.35-0.4‰, vs. PDB) at intermediate-deep water depth in
the northern Indian Ocean are interpreted as reflecting an increased contribution of better-ventilated deep water,
namely North Atlantic Deep Water (NADW), during the Holocene (e.g., Naqvi et al., 1994; Ma et al., 2019) (Fig.
S3). Although the intermediate benthic $\delta^{13}C$ record from core MD77-191 is missing for the LGM, the average
value for the Holocene (~0.31‰, vs. PDB) is consistent with previous studies carried out in the northern Indian
Ocean, and may also be associated with well-ventilated conditions (Fig. S3). The predominance of Benthic
foraminifera assemblage 2 in the early Holocene seems be in agreement with the higher values of benthic $\delta^{13}C$,
reflecting better-ventilated water masses, associated with NADW, at the core site.
By contrast, *B. aculeata* and *C. pachyderma* are major components of assemblage 1 (during the late Holocene),
together with *P. bulloides* and *E. trigona* (Figs. 3 and S2). Living *B. aculeata* have a widespread distribution,
with a preference for water depths ranging from 1500 to 2500m, and are typically associated with high organic
carbon fluxes (Mackensen et al., 1995; Almogi-Labin et al., 2000; Caulle et al., 2015). *P. bulloides* is a shallow
infaunal species, which prefers mesotrophic environments and shows adaptability with respect to oxygen
concentration in the Arabian Sea (Gupta and Thomas, 1999; Caulle et al., 2015). *E. trigona* is commonly
recorded in low oxygen habitats (Caulle et al., 2015). We thus interpret assemblage 1 as indicating relatively
low-oxygen and meso- to eutrophic bottom water conditions during the late Holocene (6-1.4 cal kyr BP).
However, the lower oxygen concentrations reflected by benthic fauna 1 seem to be the opposite of what would
be expected under an enhanced influence of better ventilated NADW during the Holocene in the northern Indian
Ocean. The higher relative abundances of *Globigerina bulloides*, a proxy of upwelling activity, observed in the
late Holocene of the same core, MD77-191, suggest increased productivity in the southeastern Arabian Sea
(Bassinot et al., 2011) (Fig. 5). This record is synchronous with the benthic foraminiferal assemblage 1 (during
the late Holocene). Thus, increased surface productivity during the late Holocene could result in more organic
matter in the bottom water, leading to depleted oxygen conditions in bottom water.
When we compare benthic assemblages 2 and 3 (during the last deglaciation and early Holocene; 17-6 cal kyr
BP) to the fauna 1 (during the late Holocene), assemblages 2 and 3 indicate that intermediate water masses were
characterized by higher bottom water oxygen conditions and a lower flux of organic matter. This is associated
with depleted *Globigerina bulloides* abundances during the same time interval compared with the late Holocene,
suggesting lower productivity in the southeastern Arabian Sea in the period from the last deglaciation to the
Holocene (Bassinot et al., 2011) (Fig. 5). Therefore, all of these elements suggest that changes in primary
productivity seem be the main factor impacting on the distribution of benthic assemblages at core MD77-191 site,
especially during the Holocene, rather than changes in intermediate-water circulation.
In addition, the total organic carbon ($C_{org}$) could also be used as a qualitative indicator of past productivity
and/or bottom water ventilation changes (Naidu et al., 1992; Canfield, 1994; Calvert et al., 1995; Naik et al.,
2017). In order to examine the relationships between intermediate $Cd_w$ and these different processes (surface
productivity and/or water mass ventilation) since the last deglaciation in the eastern Arabian Sea, we compared
the MD77-191 $Cd_w$ values with the relative abundance of *G. bulloides* and benthic foraminiferal assemblage
analyses from the same core MD77-191, together with the records for $C_{org}$ and the *G. bulloides* percentage
obtained from core SK237 GC04 (1245m, southeastern Arabian Sea, Naik et al., 2017) (Fig. 5). The MD77-191
*H. elegans* $Cd_w$ records display a strong co-variation with the $C_{org}$ from core SK237 GC04 since 17 cal kyr BP,
and are also in good agreement with the relative abundance of *G. bulloides* records during the Holocene (Fig. 5).



In addition, previous studies have suggested that increased $Cd_w$ values (>1 nmol/kg) could correspond to
elevated surface productivity (Bostock et al., 2010; Olsen et al., 2016). Thus, we suggest the intermediate $Cd_w$ at
core MD77-191 site was mainly influenced by surface productivity, especially during the Holocene.
Compared with benthic foraminifera fauna analysis from MD77-191 in the Arabian Sea, the benthic
assemblages of core MD77-176 suggest that the intermediate water masses were characterized by oligotrophic to
mesotrophic conditions and/or well-ventilated environments during the Holocene (Ma et al., 2019), associated
with much lower surface productivity (Fig. S4). This observation is in agreement with low primary productivity
during the Holocene reconstructed by the relative abundance of coccolith species *Florisphaera profunda* from
the same core MD77-176 in the northeastern BoB (Zhou et al., 2020). In the modern ocean, Prasanna Kumar et
al. (2001) indicate that primary productivity in the BoB is much lower than in the Arabian Sea, the lower surface
productivity resulting from the large freshwater input from river and direct rainfall as a result of Indian Summer
Monsoon precipitation (e.g., Vinayachandran et al., 2002; Madhupratap et al., 2003; Gauns et al., 2005).
Moreover, when we compare the average $Cd_w$ value of core MD77-176 from the BoB (~0.9 nmol/kg) with
results from core MD77-191 in the Arabian Sea (~1.2 nmol/kg), lower values, especially during the late
Holocene, are in agreement with the benthic assemblages.
To sum up, variations in the benthic assemblages seem to be associated with changes in the organic matter
flux, linked to surface productivity, especially in the Arabian Sea (Schnitker, 1994). The benthic foraminiferal
fauna are consistent with the $Cd_w$ record of core MD77-191 particularly during the late Holocene (6-1.4 cal kyr
BP). Thus, our results seem to show that the record of $Cd_w$ is mainly controlled by changes occurring at the
surface. However, during the YD, the percentages of planktonic species *G. bulloides* from cores MD77-191 and
SK237 GC04 all indicate a slight increase in paleo-productivity, the opposite of what is suggested by the results
of core MD77-191 $Cd_w$ and $C_{org}$ obtained from core SK237 GC04. This interval is also marked by enriched *G.*
*ruber* $\delta^{18}O$ values, indicating a weaker monsoon and reduced freshwater inputs (Naik et al., 2017). This apparent
discrepancy may be related to changes in the intermediate water mass sources and/or ventilation during the last
deglaciation. Therefore, we will discuss these issues in greater detail below in order to decipher these different
processes.

**5.3. Relationships between primary productivity and monsoon intensity**

During the Holocene, the intermediate $Cd_w$ records obtained from cores MD77-191 and MD77-176 display
depleted values in the early Holocene, followed by an abrupt increasing trend at the middle Holocene, and then
show a decreasing trend between 5.2 and 2.4 cal kyr BP.
Of the two cores, core MD77-176, located in the northeastern BoB, shows the lowest intermediate $Cd_w$ (down
to ~ 0.83 nmol/kg) during the 10-6 cal kyr BP time interval. Observations described above suggest that this low
in $Cd_w$ resulted from low primary productivity and thus reduced fluxes of organic matter to the intermediate
depths. We attribute this evolution to monsoon variation. The early Holocene Climate Optimum (10-6 cal kyr BP)
is characterized by enhanced monsoon precipitation (Marzin et al., 2013; Contreras-Rosales et al., 2014) (Figs.
6c-e) that resulted in increased freshwater discharge from the Ganges-Brahmaputra river system and from the
Irrawaddy River. It is likely that this increase in fresh water drove pronounced ocean stratification in the
northeast BoB, inducing low productivity.





A similar low in $Cd_w$ is observed in the reconstructed intermediate water $Cd_w$ record from core MD77-191
during the early Holocene, with values descending to ~ 0.92 nmol/kg, in the 10-6 cal kyr BP time interval. These
low values of intermediate $Cd_w$ are coeval with low surface productivity as recorded by the *G. bulloides*
percentage and low values in $C_{org}$ content from SK237 GC04 in the Arabian Sea. Off the southern tip of India,
we cannot reject the possibility that increased monsoon precipitation and enhanced freshwater runoffs in the BoB
during the early Holocene, inducing a stronger stratification, could explain part of the decrease in surface
primary productivity. Yet, at this site, another explanation prevails which is related to the decrease of summer
monsoon wind intensity that drives local Eckman pumping. As shown by Bassinot et al. (2011), the productivity
variations at the southern tip of India are inversely related to the evolution of upwelling activity along the Oman
Margin, to the west of the Arabian Sea. Based on a data/model comparison, Bassinot et al. (2011) showed that
this anti-correlation can be attributed to the northward shift of the ITCZ when boreal summer insolation reached
a maximum in the early Holocene; this ITCZ location results in enhanced summer monsoon wind intensity and
an increase in the associated Eckman pumping in the west of the Arabian Sea, and along the Oman margin, while
it weakens at the southern tip of India. This process may thus induce a decrease in surface productivity in the
southeastern Arabian Sea.
In addition, Naik et al. (2017) pointed out the co-existence of low productivity during the early Holocene in
the BoB and to the South of India, in agreement with our data that clearly show the impact of such a reduction of
surface primary productivity on the intermediate water $Cd_w$. These authors suggested a direct relationship
between intense monsoon rainfall and reduced surface productivity. However, the northeastern BoB received a
much larger amount of river input than the southern tip of India during the early Holocene (Marzin et al., 2013).
Thus, it seems reasonable to propose that the northeastern BoB is more affected by the salinity-related
stratification effect, while the southern tip of India is more affected by the decrease in wind intensity (Bassinot et
al., 2011) with enhanced stratification being potentially made stronger by an additional fresh-water effect,
although weaker than in the BoB. Ultimately, both climatic features (summer wind intensity and precipitation)
are directly under the control of monsoon evolution resulting from the orbital forcing of low latitude boreal
summer insolation.
By contrast, higher intermediate $Cd_w$ values from core MD77-191 associated with higher *G. bulloides* relative
abundances and $C_{org}$ from core SK237 GC04 during the 5.2-2.4 cal kyr BP time interval could indicate enhanced
productivity during the mid to late Holocene (Naik et al., 2017) (Fig. 5). To a lesser extent, this is also observed
in the records from the Northern BoB for the same time-period. These changes are consistent with a weakened
summer monsoon intensity, with less rainfall during the late Holocene, as observed in the BoB and over the
Indian continent (Marzin et al., 2013; Contreras-Rosales et al., 2014; Sarkar et al., 2015), and a progressive
increase in monsoon summer winds to the South of India (Bassinot et al., 2011) (Fig. 6). These observations
could also strongly support the hypothesis that the major control on surface productivity is linked to monsoon
evolution in the BoB and at the southern tip of the Arabian Sea during the Holocene (Bassinot et al., 2011; Naik
et al., 2017; Zhou et al., 2020).

**5.4. Millennial-scale changes in intermediate water circulation during the deglaciation**

During the last deglaciation, short events have been recorded at the site of core MD77-191 during the 16-15.2

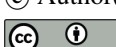



(HS1) and 12.6-11 cal kyr BP (YD) time intervals (Fig. 5). The low $Cd_w$ values in the MD77-191 record are
coeval with reductions of $C_{org}$ in core SK237 GC04 during the HS1 and YD periods (Fig. 5). According to
previous studies, extremely high $Cd_w$ values (>1 nmol/kg) were reported to have been associated with enhanced
surface productivity (Bostock et al., 2010; Olsen et al., 2016). However, the range of values of intermediate $Cd_w$
(0.58-0.85 nmol/kg, HS1; 0.5-0.8 nmol/kg, YD) from core MD77-191 during the last deglaciation is much lower
compared with the Holocene $Cd_w$ values (>1 nmol/kg), and thus may be associated with other processes such as
a better ventilation, changes in the water mass source, and/or depleted surface productivity (Fig. 7). Significant
decreases in *G. bulloides* relative abundance of cores SK237 GC04 (Naik et al., 2017) and MD77-191 records
were observed from the HS1 to B-A (Bassinot et al., 2011) (Fig. 5). Although a slight increase occurred in the
YD, the *G. bulloides* percentage records from both cores show a general depletion during the last deglaciation
compared with the last glacial interval and late Holocene (Fig. 5). Thus, we do not expect that surface
productivity played an important role during the last deglaciation. In addition, compared with the relative
percentage of *G. bulloides* during the B-A, slightly higher values at both core sites during the HS1 and YD may
indicate a small, but net increase of surface productivity during these intervals (Fig. 5). This should have led to
increased intermediate $Cd_w$ and organic matter preservation under conditions of low oxygen concentration. But,
we observe a decrease in these two proxies, the opposite of what would be expected from stronger surface
productivity. Thus, this apparent discrepancy provides evidence for the influence of changes in water masses
and/or ventilation during the HS1 and YD.

Moreover, an increase in benthic $\delta^{13}C$ values is observed during the HS1 and YD in the northern Indian Ocean

(e.g., Duplessy et al., 1984; Curry et al., 1988; Naqvi et al., 1994; Jung et al., 2009; Ma et al., 2019). The
increase in the different benthic $\delta^{13}C$ records during the HS1 and YD in the western Arabian Sea, Pacific Ocean
and BoB is interpreted as reflecting the northward expansion of AAIW (Pahnke and Zahn, 2005; Jung et al.,
2009; Ma et al., 2019). The decreased benthic-planktonic foraminiferal $^{14}C$ offset (B-P age) obtained from
marine sediment cores from the Arabian Sea and the Bay of Bengal during the same intervals could confirm
enhanced vertical mixing in the Southern Ocean (Bryan et al., 2010; Ma et al., 2019). The transition in the $\varepsilon_{Nd}$
and $\Delta^{14}C$ records during the deglaciation also indicates a strong northward penetration of AAIW within the
North Atlantic and Bay of Bengal (e.g., Cao et al., 2007; Pahnke et al., 2008; Pena et al., 2013; Yu et al., 2018).
In addition, during the HS1 and YD, a decrease in the $[CO_3^{2-}]$ record from core MD77-191 also suggests the
release of $CO_2$ from the deep ocean in the deglacial period through the expansion of AAIW (Ma et al., 2020).
These time intervals are associated with better ventilation in the Southern Ocean (e.g., Anderson et al., 2009;
Skinner et al., 2010), which led to enhanced vertical ventilation resulting in increased production of intermediate
water masses (AAIW) (Anderson et al., 2009).

As mentioned before, previous studies have suggested an enhanced northward flow of southern sourced

intermediate water mass AAIW both in the Atlantic, Pacific and Indian Oceans during the last deglaciation (e.g.,
Pahnke et al., 2008; Bryan et al., 2010; Poggemann et al., 2017; Yu et al., 2018; Ma et al., 2019, 2020),
indicating that the source of intermediate water masses may be partly the same in these oceans. Thus, by using
the benthic $\delta^{13}C$ values collected from the north Indian Ocean to better constrain the influence of AAIW in the
two studied cores (Naqvi et al., 1994; Jung et al., 2009; Ma et al, 2019; 2020), we can also compare the range
values of AAIW $Cd_w$ obtained from the cores MD77-191 and MD77-176 with other oceans, including data from
the Atlantic and Pacific Oceans at intermediate water depth during the HS1 and YD ($Cd_w$, 0.3-0.9 nmol/kg;





Umling et al., 2018; Valley et al., 2017). Unfortunately, the resolution of both intermediate $Cd_w$ and benthic $\delta^{13}C$
from core MD77-176 (northeastern BoB) are very low for the HS1 and YD events, making it difficult to extract
reliable information. Thus, we have decided to focus on the results from core MD77-191 (0.5-0.85 nmol/kg)
during these two time-intervals; these results are in good agreement with the collected dataset (Fig. 7). Thus, the
benthic $Cd_w$ results provide new evidence for tracking the northern flow of AAIW in the northern Indian Ocean,
which increased during HS1 and the YD.
Taken together, $Cd_w$, B-P age offset, benthic $\delta^{13}C$, $\varepsilon_{Nd}$ and $\Delta^{14}C$ records reported from the northern Indian
Ocean all suggest strong upwelling and enhanced northern flow of AAIW from the Southern Ocean during HS1
and the YD. Thus, the variations in these records can provide strong evidence for the hypothesis that Southern
Ocean upwelling played a vital role in the icrease of atmospheric $CO_2$ in the deglacial period (Anderson et al.,
2009; Skinner et al., 2010, 2014). However, Kohfeld et al. (2005) suggested that although physical processes
(such as ventilation) are involved in the glacial-interglacial atmospheric $CO_2$ change, the biological pump may
also contribute nearly half of the observed changes of $CO_2$ during the glacial-interglacial transitions. As shown
above, the HS1 event is characterized by reduced surface productivity, as revealed by the lower percentage
values of *G. bulloides* in core MD77-191 (Bassinot et al., 2011) and by several studies of cores located in the
eastern and western Arabian Sea within the Oxygen Minimum Zone (e.g., Schulz et al., 1998; Altabet et al.,
2002; Ivanochko et al., 2005; Singh et al., 2006, 2011; Naik et al., 2017). This reduced productivity at a
millennial timescale suggests that the entire biological factory was related to the reduced monsoon intensity
during the North Atlantic Heinrich events (e.g., Singh et al., 2011; Naik et al., 2017). Thus, a weaker biological
production could also have contributed to the two-step increase of atmospheric $CO_2$ during the last deglaciation,
at least for the HS1 period.

**6. Conclusions**

Changes in benthic foraminiferal Cd/Ca and assemblages were reconstructed on core MD77-191 (1254 m
water depth) located off the southern tip of India, as well as on core MD77-176 (1375 m water depth) from the
northern BoB, in order to reveal the evolution of intermediate water circulation and paleo-nutrient changes in the
northern Indian Ocean since the last deglaciation. We reconstructed seawater $Cd_w$ concentration by converting *H.*
*elegans* Cd/Ca. Benthic Cd/Ca ratios are mainly influenced by changes in surface productivity and intermediate-
bottom water ventilation.
Results indicate that assemblages 2 and 3, reflecting high bottom water oxygen conditions and a low flux of
organic matter, dominated between 17 and 6 cal kyr BP, corresponding to a poor productivity time-period. The
typical late Holocene assemblage indicates a relatively low-oxygen level and meso- to eutrophic deep-water
conditions, associated with high surface productivity. The early Holocene (10-6 cal ka BP) corresponds to a low
in productivity associated with depleted $Cd_w$ in intermediate water. These observations seem to result from
enhanced monsoon precipitation and increased river inputs from the Himalayan Rivers, which led to more
marked stratification in the BoB and a reduction in primary and export productivity. At the southern tip of India,
the decrease in vertical mixing is also associated with a reduction in summer wind forcing resulting from the
northward displacement of ITCZ during summer (Bassinot et al., 2011). During the late Holocene (5.2-2.4 cal
kyr BP), the increased intermediate $Cd_w$ concentrations of cores MD77-191 and MD77-176 indicate enhanced
surface productivity in the southeastern Arabian Sea and in the northeastern BoB, corresponding to weakened



monsoon intensity and rainfall, in agreement with other local records and reconstructions of the paleo-monsoon
strength. Thus, our results clearly show the strong control of intermediate water $Cd_w$ during the Holocene by
orbitally-driven changes in summer monsoon productivity.
As far as millennial-scale variability is concerned, during the last deglaciation, decreased intermediate $Cd_w$
concentrations during HS1 and the YD are coeval with increased benthic $\delta^{13}C$, depletion in $[CO_3^{2-}]$ and
decreased B-P age offsets. These observations indicate that the low $Cd_w$ values in intermediate water mainly
resulted from the increased northward flow of AAIW during HS1 and YD intervals. These signals also provide
strong evidence for the important role of enhanced Southern Ocean ventilation in the $CO_2$ increase during the
last deglaciation. The declined intermediate $Cd_w$ obtained from southeastern Arabian Sea (Core MD77-191),
combined with the published eastern and western Arabian Sea paleo-productivity results, together provide
evidence for the important influence of decreased monsoon intensity at a millennial time scale during cold events
in the North Atlantic region, associated with the increase in atmospheric $CO_2$ during the last deglaciation.

**Data availability**
All data are given in Table 1 and supplementary materials Tables S1-S2.

**Supplement**
The supplement related to this paper is available online.

**Author contribution**
RM, SS, FB and CC developed the idea and interpreted the results. CC and FB supplied foraminifera samples.
RM did benthic foraminifera assemblage and geochemical analyses with the aide of FH and LL. ZY and LL
joined the discussion. All co-authors helped to improve the article.

**Competing interests**
The authors declare that they have no conflict of interest.

**Acknowledgements**
R. Ma gratefully acknowledges the China Scholarship Council for providing funding for her study in France.
The research leading to this paper was funded by the French National Research Agency under the
"*Investissements d'avenir*" programme (Grant ANR-11-IDEX-0004-17-EURE-0006), and the INSU-LEFE-
IMAGO-CITRON GLACE project.

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





**Table 1.** Species composition of benthic foraminiferal assemblages from core MD77-191.

| | Dominant species | | Important associated species | | Variance (%) |
|---|---|---|---|---|---|
| PC1 | | | | | 42 |
| Positive loadings | *Bulimina aculeata* | 0.84 | *Pullenia bulloides* | 0.18 | |
| | *Cibicidoides pachyderma* | 0.19 | *Ehrenbergina trigona* | 0.13 | |
| Negative loadings | *Hoeglundina elegans* | -0.14 | *Cibicidoides wuellerstorfi* | -0.04 | |
| | *Bulimina manginata* | -0.07 | *Globocassidulina subglobosa* | -0.06 | |
| PC2 | | | | | 19 |
| Positive loadings | *Sphaeroidina bulloides* | 0.42 | *Gyroidinoides orbicularis* | 0.17 | |
| | *Bulimina mexicana* | 0.11 | *Gyroidinoides soldanii* | 0.07 | |
| Negative loadings | *Bulimina aculeata* | -0.14 | *Hoeglundina elegans* | -0.62 | |
| | *Cibicidoides pachyderma* | -0.07 | | | |





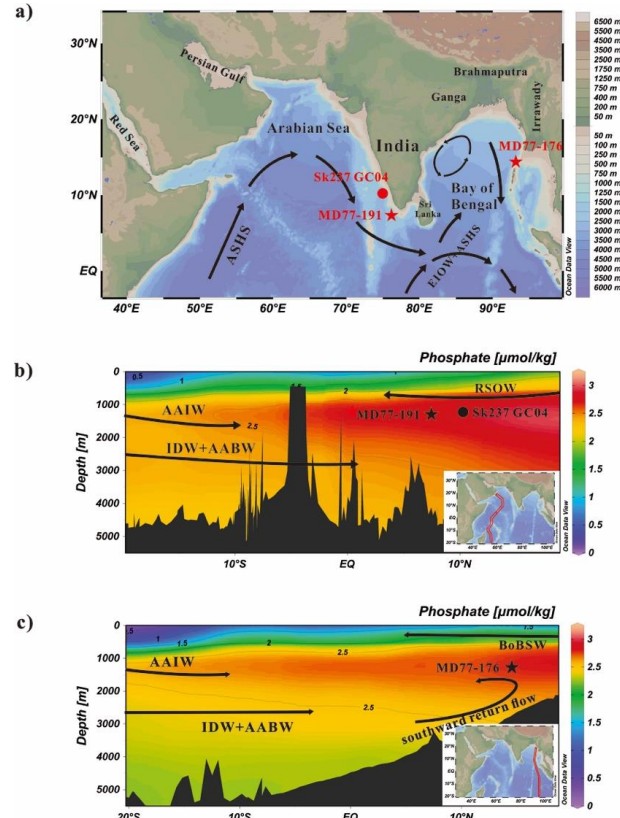

**Fig. 1.** (a) Oceanographic setting and locations of core MD77-191 in the Arabian Sea (red star), core MD77-176 in the Bay of Bengal (red star) and reference site SK237 GC04 (red circle, Naik et al., 2017). The black arrows represent the general surface circulation direction in the Northern Indian Ocean during the summer, Southwest Monsoon (Schott and McCreary, 2001). (b) and (c) Phosphate distribution along depth-latitude sections during the Southwest Monsoon and Northeast Monsoon periods, for the Arabian Sea and the Bay of Bengal, respectively. Data (in µmol/kg, colored scale) were contoured and plotted using the Ocean Data View (ODV) software (Schlitzer, 2015). On these two figures are shown the distribution and circulation of water masses in the Arabian Sea and Bay of Bengal (black arrows). ASHS: Arabian Sea High Salinity Water, EIOW: Eastern Indian Ocean Water, BoBSW: Bay of Bengal surface waters, AAIW: Antarctic Intermediate Water, RSOW: Red Sea Overflow Water, AABW: Antarctic Bottom Water, IDW: Indian Deep Water.

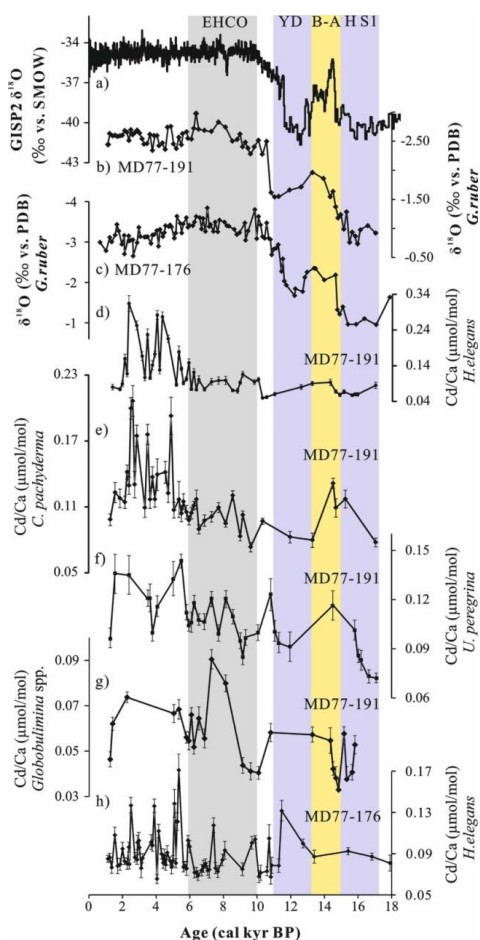

**Fig. 2.** (a) GISP2 Greenland ice core $\delta^{18}$O signal (Stuiver and Grootes, 2000). (b)-(c) *Globigerinoides ruber* $\delta^{18}$O records of cores MD77-191and MD77-176, respectively (Marzin et al., 2013; Ma et al., 2020). (d)-(g) Cd/Ca records of the benthic foraminifera *Hoeglundina elegans*, *Cibicidoides pachyderma*, *Uvigerina peregrina*, and *Globobulimina* spp. obtained from core MD77-191; (h) Cd/Ca records of the benthic foraminifera *H. elegans* from core MD77-176. EHCO for Early Holocene Climate Optimum, YD for Younger Dryas, B-A for Bølling-Allerød and HS1 for Heinrich stadial 1.



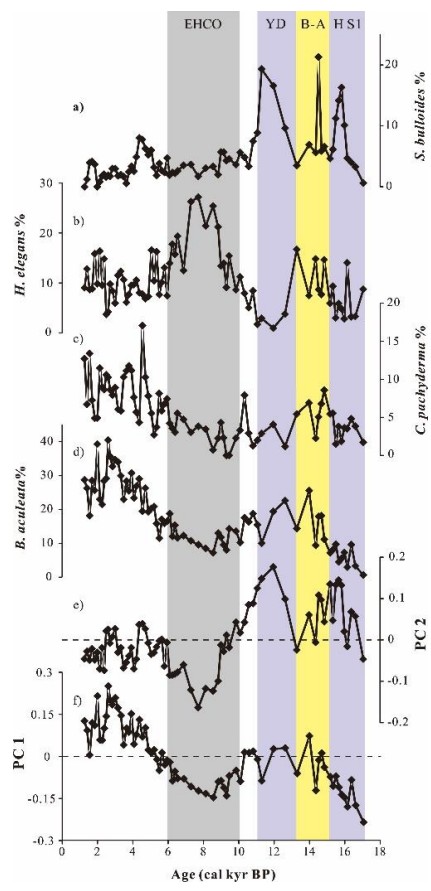

**Fig. 3.** Down core variations of PC scores and the percentages of major species. The color shaded intervals and
abbreviations are the same as in Figure 2.





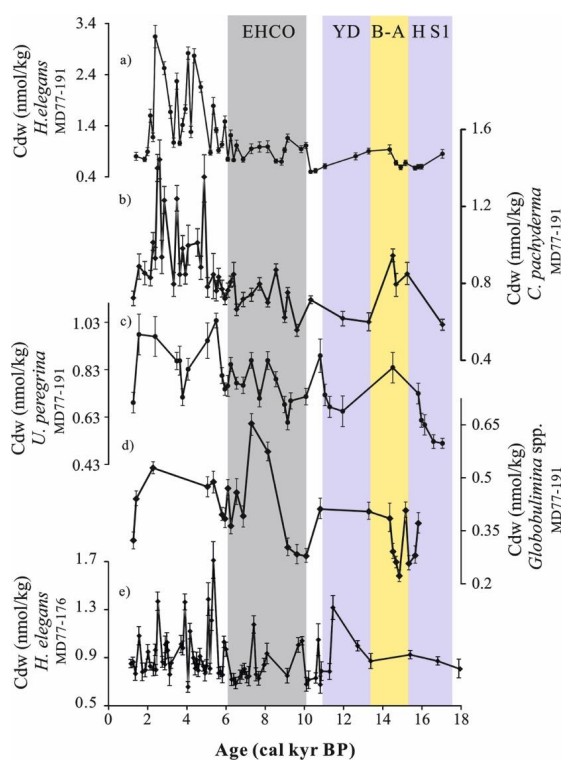



**Fig. 4.** (a)-(d) Cd$_w$ records calculated based on the Cd/Ca of benthic foraminifera *Hoeglundina elegans*,
*Cibicidoides pachyderma*, *Uvigerina peregrina*, and *Globobulimina* spp. obtained from core MD77-191, (e) Cd$_w$
record from core MD77-176 reconstructed using *H. elegans* Cd/Ca. The color shaded intervals and abbreviations
are the same as in Figure 2.





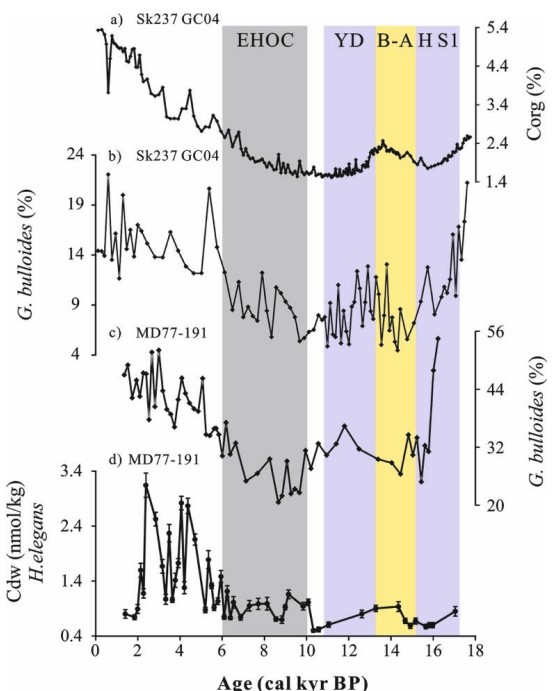



**Fig. 5.** (a) Organic carbon weight percentage (%$C_{org}$) and (b) *G. bulloides* percentage from core SK237 GC04 (1245m, Arabian Sea, Naik et al., 2017). (c) Relative abundance of *G. bulloides* (Mléneck, 1997; Bassinot et al., 2011) and (d) $Cd_w$ records from core MD77-191 (Arabian Sea). The color shaded intervals and abbreviations are the same as in Figure 2.



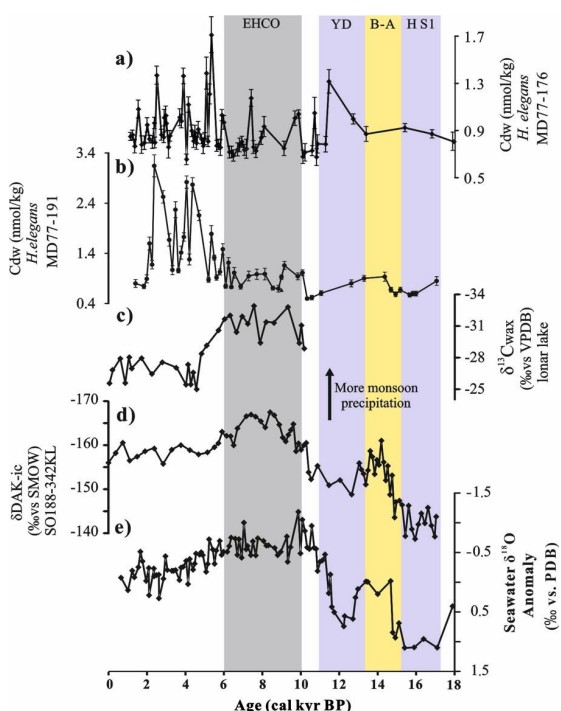

**Fig. 6.** (a) and (b) intermediate $Cd_w$ calculated from *H. elegans* obtained from MD77-176 and MD77-191, respectively. (c) Lonar Lake $\delta^{13}C_{wax}$ record (Sarkar et al., 2015). (d) $\delta DAk_{-ic}$ record from core SO188-342KL (Contreras-Rosales et al., 2014). (e) Seawater $\delta^{18}O$ anomaly obtained from MD77-176 (Marzin et al., 2013). The color shaded intervals and abbreviations are the same as in Figure 2.





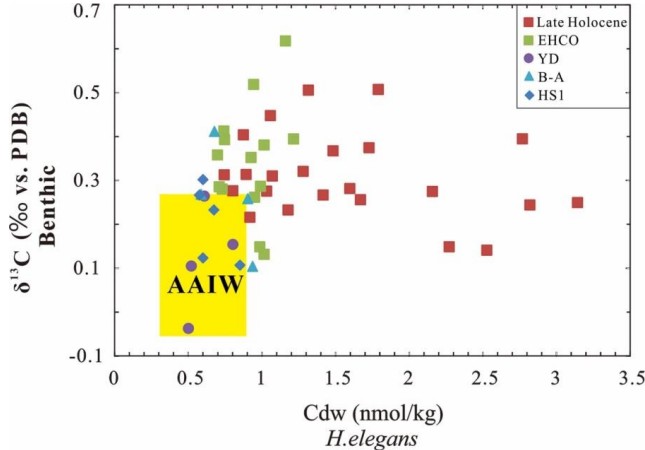

**Fig. 7.** Intermediate $Cd_w$ versus benthic $\delta^{13}C$ obtained from core MD77-191 located off the southern tip of India.
The yellow shaded area represents the ranges of $Cdw$-$\delta^{13}C$ values of AAIW during the HS1 and YD, which were
reconstructed in the Indian Ocean (benthic $\delta^{13}C$, Naqvi et al., 1994; Jung et al., 2009; Ma et al, 2019; 2020),
Pacific and Atlantic Oceans (benthic $Cd_w$, Valley et al., 2017; Umling et al., 2018) at intermediate water depths.
The abbreviations are the same as in Figure 2.