# Peer review of "Changes in productivity and intermediate circulation in the northern Indian Ocean since the last deglaciation: new insights from benthic foraminiferal Cd/Ca records and benthic assemblage analyses"

_Climate of the Past, 2020_

## Referee Comment (RC1) · Anonymous Referee #1 · 1 Apr 2021

Review of manuscript "Changes in productivity and intermediate circulation in the northern Indian Ocean since the last deglaciation: new insights from benthic foraminiferal Cd/Ca records and benthic assemblage analyses" by Ma et al.. This manuscript presents data from sediment cores from the northern Indian Ocean (Arabian Sea and Bay of Bengal), comprising geochemical time series generated on benthic foraminifera as well as census data for planktic and benthic foraminifera. Based on these data, the authors deduce changes in monsoon driven changes in productivity

mainly dominating the various records during the Holocene and changes in intermediate water chemistry during the deglaciation. In principle the authors present interesting data and some of the interpretations appear justified. There are, however, a number minor and more major issues, preventing recommending publication as is at this stage. These are some of the issues:

a) The biggest issue is related to the lack of constituency of interpreting the Cdw records. In lines 450-453 the authors claim that the Cdw values during the deglaciation are lower than during the Holocene. First, this statement is only correct if longer term averages are considered. On short time scales (which need to be considered, given that this is a chapter on millennial scale change), the youngest Cdw data in core MD77-191 (2-1.5 KaBP) are comparable to YD and HS1 values. Up to this point a big effort has gone into establishing Cdw as reflecting productivity variations at the sea surface and the related flux of organic carbon. Now the focus shifts to bottom water ventilation changes being recorded. If general water ventilation would play are role in setting the recorded Cdw values, this has to apply to the Holocene too and would therefore need to be considered there too. Interestingly, the authors do involve water ventilation during the Holocene in relation to the carbon isotope and census data, but not very much in relation to the Cdw records. Also, if the general interpretation for the Holocene section is used, why is there no change in the Cdw record around 16-16.5 KaBP? During this time, high G. bulloides concentrations (highest in the entire MD77-191 record) in the same core are shown in figure 5. High concentrations of G. bulloides strongly support the notion of enhanced productivity, as the authors themselves assume in case of the Holocene changes G. bulloides concentrations. Around 16-16.5 KaBP the high G. bulloides concentrations are not reflected in the Cdw data. This would suggest that the Cdw water are not very reflective of surface productivity changes, casting doubts on parts of the Holocene storyline. This would need to be addressed in a revised version, not only in this section but in large parts of the manuscript.

b) Also, in line 329 (and thereafter), the authors, for the first time, mention NADW,

claiming that this water mass would dominate during the early Holocene at site MD 77-191. How does this claim compare to the modern water mass distribution in the area? Is it not true that most of the deepwaters in the Indian Ocean are mixes of various endmembers, of which NADW is just one? The only place original (largely unmixed) NADW occurs is off the southeast coast of Africa, with the northward propagation blocked by the Davie Ridge (although there is some discussion in relation to a potential northward spillover occurring). In order to substantiate their argument, a) the hydrography section needs improvement and b) there needs to be a more in-depth explanation how (even contributions) of a deepwater mass, currently occurring below ~2km in in the Mozambique channel, affect sediment cores at true intermediate water depth. The latter changes affect the discussion of the entire Holocene record.

c) There is some inconsistency regarding the description (interpretation?) of the habitat of the various benthic foraminifera species used in the study. In lines 141 and 142, the authors state that C. pachyderma is an epifaunal species. In contrast, in 289 and 290 they state that it is a shallow infaunal species. This needs to be clarified and consistently used throughout the manuscript.

d) At times the description of results/findings is too generic. As an example in lines 364 and following, a number of comparisons are made regarding the similarity of records. Generally, on longer time scales, yes there is some similarity. It should be pointed out though that there are also substantial differences at the millennial scale. This is particularly relevant for the comparison between Corg and H. elegans Cdw records. This needs a better wording.

e) (minor point) Figure 6 needs a better embedding/explanation in the manuscript. Some of the records are neither explained in the main text nor in the figure caption.

Overall, there are some useful data in this manuscript. The discussion of the data and subsequent interpretation lacks maturity at this stage and requires improvement. A moderate to major revision is required.

---

## Referee Comment (RC2) · André Bahr (Referee) · 4 Apr 2021

**Review of *Changes in productivity and intermediate circulation in the northern Indian Ocean since the last deglaciation: new insights from benthic foraminiferal Cd/Ca records and benthic assemblage analyses* by Ma et al.**

The authors present benthic foraminiferal assemblage records and Cd/Ca data from the western Arabian Sea and Bay of Bengal (BoB) to investigate surface primary productivity (PP) and intermediate water mass variability in the context of the last deglaciation and Holocene climatic evolution. The authors find that Cd/Ca is primarily controlled by PP during the Holocene which mirrors monsoonal intensity. Notably, a strong monsoon is inferred to suppress PP in both areas due to enhanced run-off which increases upper ocean stratification in the BoB and reduced Ekman-upwelling off India as a result of decreased wind stress. During the deglaciation, the authors infer a dominance of water mass changes in driving the Cd/Ca signal, showing an enhanced advection of AAIW into the northern Indian Ocean during YD and HS1.

In general, the data and its interpretation appear mostly sound and in line with existing concepts about the paleoceanography of the Arabian Sea and the BoB as well as the influence of the monsoon on the PP in these areas. In this respect I find it noteworthy that the data (i) supports the presumed E-W dipole between strong upwelling off Oman and weak upwelling off India, and (ii) that maximum monsoon-induced run-off in the BoB apparently suppresses PP due to strong stratification, despite riverine nutrient input should enhance plankton blooms on surface level. The authors mention stratification as an explanation rather briefly, however, I would encourage the authors to devote one or two more sentences on this issue (see also my detailed comment).

I can also follow the arguments for the inferred intrusion of AAIW into the northern Indian Ocean during HS1 and YD, which agrees with the well-documented enhanced northward protrusion of this water mass in the Atlantic Ocean. However, I am not convinced by the way the authors come to this conclusion, which is based on the claimed mismatch between increasing *G. bulloides* abundances and decreasing Cdw estimates during YD and HS1. As depicted in the figure below, both records essentially follow the same trend, also bearing in mind that the resolution of Cdw is relatively low during YD and HS1. Hence, the proposed anti-correlation between *G. bulloides* abundances and Cdw seems to be an overstatement.

Irrespective of this problem, the good match of Cdw and d13C with AAIW reference records make it reasonable to assume that the Cdw values in deed capture water mass variability between HS1 and YD (Fig. 7). Hence, the interpretation at the end seems correct, but it is more likely that the relatively modest increase in PP during the YD/HS1 appear to have had a negligible influence of the Cdw. Only if PP is really high (such as in the mid-Holocene) Cdw is dominated by PP, as also indicated by the very high values > 1.0. The authors are somewhat over-confident regarding the use of Cd/Ca as a water mass tracer in the such potentially highly productive areas and might consider toning down their argumentation.

While the manuscript is well written, the Figures might benefit from rearrangement to make the discussion more easier to follow (cf. detailed comments below).

Given some moderate revisions I support publication of this study which represent an important contribution to our understanding of the deglacial evolution of the Indian Ocean.

[Figure]

Figure 5 with Cd/Ca from MD77-191 in blue, illustrating that Cd/Ca follows *G. bulloides*-abundances (even more in the more high-resolution data of adjacent core GC04).

**Detailed comments**

Line 78: The motivation for the study is rather weak (essentially: we know little about the paleoproductivity of the BoB). It would be good to more explicitly state why we should care about this issue.

L. 85: "estimate past changes in the nutrient content, since the last deglaciation, over the last 17 kyr BP." The last part is redundant.

l. 100: "of the planktonic…"

l. 101 (and elsewhere): avoid using "." as multiplicator

L. 109: "Arabian High Salinity Waters" (all capitals)

L. 118: Neither Fig.1 nor S1 show salinity.

L. 131: "northern intermediate …" (no capitals)

Section 3: Please also include the statistical methods used in the study in the Methods chapter. Which program did you use to perform the PCA? Did you use a correlation or variance/covariance matrix?

Section 3.1.: Regarding the design of the study, I wonder why the authors decide to use four different species, when *H. elegans* is available as a well-documented, faithful recorder of bottom water Cd/Ca. What was the rationale to use the three calcitic species, especially as they include infaunal dwellers which are naturally not the best suited for detecting bottom water fluctuations?

L. 204-205: you might omit "over the last deglaciation"; add an "a" before "significant decrease"

L. 228 etc., regarding the PCA results: You show 2 PCs which explain 61% of the total variance. What's about the other PCs, how much variance to they explain and what was the rationale to limit the investigations to those two PCs? When the authors discuss the loadings of the individual PCs, they assign particular species with a very low loading to individual assemblages. PC1 for example is very much dominated by *B. aculeata* (+0.84 loading); the denoted loadings of -0.07 and less for *B. manginata, C. wuellerstorfi, G. subglobosa* (Table 1) appear to be rather insignificant. The same applies for PC2 which has high loadings of +0.42 and -0.62 for *S. bulloides* and *H. elegans*, respectively; I doubt that e.g. *G. soldanii* with a loading of 0.07 has a significant relevance to PC 2. Please reconsider the discussion of the PC 1 and PC 2 accordingly. You might also consider providing a bi-plot for PC 1 and PC 2 as an extra figure.

L. 249: "aragonite": change into "argonitic"

L. 262: The paragraphs discussing Cdw repeat in large parts what have been written about the Cd/Ca ratio in the Results chapter. Please avoid such duplication. The same also applies for Figures 2 and 4. Figure 2 might be moved into the supplement.

L. 264: please add the modern Cdw to figure 4 for reference.

Figures 2 and 4: The y-axes should have a common scale to enable direct comparison of the individual records. As shown, the Cd/Ca and Cdw records of Globobulimina spp. appear to show large fluctuations, however, compared to the other species these fluctuations are of rather minor importance (as stated in the text).

L. 286: the core tops values mentioned here should be shown in the respective Figure for reference.

L. 300-303: The final statement that *H. elegans* provides the most reliable Cd/Ca (or Cdw) data is not really surprising. As stated in the earlier comment I would appreciate if the authors could provide more arguments what they wanted to test/proof with including the other three species.

L. 307 etc.: I think the presentation of the Assemblage data could be improved: 1) as stated earlier the presentation of the PCA results is not totally convincing; 2) why do the authors start with Assemblage 3 not in the numerical order? 3) If they use the Assemblages as environmental indicator they could plot the abundance of the respective Assemblages instead of individual foraminifera species in the results figure; they could also assign specific environmental parameters to each Assemblage in the Figures (e.g. Assemblage 1 = more/less productivity or oxygenation), which would help to more concisely convey the message of the study.

L. 351: better write "Assemblage 1" instead of "fauna 1"

L. 353. "depleted *Globigerina bulloides* abundances" – replace by "low *G. bulloides* abundances"

L. 370: please refer more often to the respective Figures.

L. 398: a decreasing Cdw trend between 5.2 and 2.4 cal kyr BP is not evident for me, the values are constantly very high during this time period.

L. 405-406: the relation of stratification and PP should be discussed in more detail (see my general remark).

L. 431-432: The authors refer to summer insolation – pleas show it in an appropriate figure (e.g. Fig. 5)

L. 453: The reference to Figure 7: should't it be rather Fig. 6?

L. 457-458: "Thus, we do not expect that surface productivity played an important role during the last deglaciation." This statement is odd, as it has been discussed at great length that PP is influencing Cd/Ca. The following "In addition,…" does also not fit as the following sentence does not support the above notion of PP playing an unimportant role.

L. 462 etc: I am not convinced by the statement that increasing *G. bulloides* abundances during HS1 and YD are in conflict with both, Corg and Cdw records. With regards to Corg I agree that it declines opposite to the trend in *G. bulloides*, however, Corg does not only depend on PP but also on preservation, and potentially sedimentation rate (one way to check the influence of sediment accumulation would be to compute Corg accumulation rates). However, Cdw rather follows *G. bulloides* abundances, at least it is not anti-correlated, as one might infer from the text.

L. 482-487: the sentence is too long and complicated, please rephrase.

L. 496: "increase" instead of "icrease"

L. 504: "the entire biological factory was related to reduced monsoon intensity" – see my general comment: this statement need more justification. The presence/absence of *G. bulloides* might well be influenced by stratification and surface water freshening, but does this apply to other primary producers as well?

**General remarks to the figures:**

It helps the reader if the authors state next to the core name where the core is located (e.g. within the BoB)

Fig. 5: you might add here the benthic d13C records of MD77-191 and -176 used to discussed water mass variability (cf. Fig. S3)

S1: please add the variable + unit to the color shading on the right side of the maps.

---

## Author Comment (AC1) · 26 Jul 2021

**Reply to reviewer #1 comments:**

We thank Reviewer 1 for providing helpful comments on our manuscript, they have been carefully considered. Please find below our answers to these comments.

Review of manuscript "Changes in productivity and intermediate circulation in the northern Indian Ocean since the last deglaciation: new insights from benthic foraminiferal Cd/Ca records and benthic assemblage analyses" by Ma et al.. This manuscript presents data from sediment cores from the northern Indian Ocean (Arabian Sea and Bay of Bengal), comprising geochemical time series generated on benthic foraminifera as well as census data for planktic and benthic foraminifera. Based on these data, the authors deduce changes in monsoon driven changes in productivity mainly dominating the various records during the Holocene and changes in intermediate water chemistry during the deglaciation. In principle the authors present interesting data and some of the interpretations appear justified. There are, however, a number minor and more major issues, preventing recommending publication as is at this stage. These are some of the issues:

a) The biggest issue is related to the lack of constituency of interpreting the Cdw records. In lines 450-453 the authors claim that the Cdw values during the deglaciation are lower than during the Holocene. First, this statement is only correct if longer term averages are considered. On short time scales (which need to be considered, given that this is a chapter on millennial scale change), the youngest Cdw data in core MD77-191 (2-1.5 Ka BP) are comparable to YD and HS1 values. Up to this point a big effort has gone into establishing Cdw as reflecting productivity variations at the sea surface and the related flux of organic carbon. Now the focus shifts to bottom water ventilation changes being recorded. If general water ventilation would play are role in setting the recorded Cdw values, this has to apply to the Holocene too and would therefore need to be considered there too. Interestingly, the authors do involve water ventilation during the Holocene in relation to the carbon isotope and census data, but not very much in relation to the Cdw records. Also, if the general interpretation for the Holocene section is used, why is there no change in the Cdw record around 16-16.5 Ka BP? During this time, high *G. bulloides* concentrations (highest in the entire MD77-191 record) in the same core are shown in figure 5. High concentrations of *G. bulloides* strongly support the notion of enhanced productivity, as the authors themselves assume in case of the Holocene changes *G. bulloides* concentrations. Around 16-16.5 Ka BP the high *G. bulloides* concentrations are not reflected in the Cdw data. This would suggest that the Cdw water are not very reflective of surface productivity changes, casting doubts on parts of the Holocene storyline. This would need to be addressed in a revised version, not only in this section but in large parts of the manuscript.

Answer: The two main comments made by the first reviewer concerns i) the influence of surface productivity changes on the intermediate $Cd_w$ records at millennial time scale during the last deglaciation (16-16.5 cal kyr BP) and the Holocene (2-1.5 cal kyr BP) and ii) the contribution of intermediate water circulation variations, especially during the Holocene. We fully agree with the reviewer that there may be some questions interpreting the $Cd_w$ records, especially at the millennial scale. Changes in intermediate $Cd_w$ values of benthic foraminifera can be influenced by different processes such as surface productivity, changes of the water mass sources and/or

ventilation (e.g., Came et al., 2008; Bostock et al., 2010; Olsen et al., 2016; Poggemann et al., 2017; Yu et al., 2019). First, at the Arabian Sea site, we suggest that the observed significant increase of intermediate $Cd_w$ values from the last deglaciation (~0.7 nmol/kg) to the late Holocene (~1.59 nmol/kg) could be associated with the surface productivity; indeed, this is supported by the *G. bulloides* record from the same core and another one close to the studied site (Bassinot et al., 2011; Naik et al., 2017), as well as by previous studies suggesting that increased $Cd_w$ values (>1 nmol/kg) could correspond to elevated surface productivity (Bostock et al., 2010; Olsen et al., 2016).

However, we also agree with the reviewer that there is some mismatch between the increased *G. bulloides* abundances and the decreased $Cd_w$ values obtained from the same core MD77-191 during the last deglaciation, especially at around 16-16.5 cal kyr BP. As the resolution of both records from core MD77-191 is relatively low during YD and HS1, it seems more reasonable to use the high resolution *G. bulloides* abundances records from the near core site SK237 GC04 (1245m, southeastern Arabian Sea, Naik et al., 2017), reflecting the surface productivity changes in this area. Therefore, we observe that although the *G. bulloides* abundances from the Arabian Sea display an increasing trend during HS1 and YD events, these modest increases in surface productivity are synchronous with low $Cd_w$ values (～0.6 nmol/kg) during HS1 and YD, much lower compared to the $Cd_w$ results during the late Holocene (~1.59 nmol/kg). Thus, we suggest that the variations of surface productivity could not be the main control at this studied site during the last deglaciation, and thus we proposed that changes in the circulation can explain the observed results. Indeed, the influence of intermediate water masses variations has also been demonstrated by many proxies ($\varepsilon_{Nd}$, benthic $\delta^{13}C$, B-P age offsets and $CO_3^{2-}$) in the northern Indian Ocean (Bryan et al., 2010; Yu et al., 2018; Ma et al., 2019; 2020). These previous studies have been summarized and detailed discussed in the manuscript (lines 465-507). Consequently, the conclusion that the influence of changes in water masses and/or ventilation on $Cd_w$ records from the northern Indian Ocean during the HS1 and YD may be still reasonable. In order to take this comment into account, we modified the discussion in section 5.4 as below:

Significant decreases in *G. bulloides* relative abundance of cores SK237 GC04 (Naik et al., 2017) and MD77-191 records were observed from the HS1 to B-A (Bassinot et al., 2011), and thereafter slight increases occurred in the YD (Fig. 5). These high values at both core sites during the HS1 and YD may indicate an enhanced surface productivity during these intervals (Fig. 5). This should have led to increased intermediate $Cd_w$ and organic matter preservation under low oxygen concentration conditions during the HS1 and YD. However, despite a low resolution for the MD77-191 $Cd_w$ record during the last deglaciation, we do not observe high values of intermediate $Cd_w$ during the HS1 and YD (～0.6 nmol/kg) compared with the late Holocene (~1.59 nmol/kg), especially at 16.5-16 cal kyr BP. Although we cannot fully discard the influence of surface productivity on the intermediate $Cd_w$ in these time intervals, this apparent discrepancy seem to provide another evidence for the influence of changes in water masses and/or ventilation during the HS1 and YD, as already demonstrated by previous studies and proxies in the northern Indian Ocean (Bryan et al., 2010; Yu et al., 2018; Ma et al., 2019; 2020).

We also would like to thank the reviewer to point out that the influence of water mass ventilation on the intermediate $Cd_w$ during the Holocene should be discussed. In order to clarify this point, we added some new parts in Section 5.2 about the possible contribution of past water masses changes to the $Cd_w$ records obtained from northern Indian Ocean during the Holocene.

Briefly, increased benthic $\delta^{13}C$ values and B-P age offsets, as well as depleted $\varepsilon_{Nd}$, $\delta^{18}O_{ivc}$ and Mg/Li results obtained from MD77-176 and MD77-191, suggest the increased influence of NADW (Yu et al., 2018; Ma et al., 2019; 2020); indeed, deep-water masses can contribute to intermediate water masses in the Northern Indian Ocean by upwelling when flowing northward (Talley et al., 2011; Naqvi et al., 1994). NADW is characterized by the fresh, well-ventilated and depleted nutrient (modern $Cd_w$, ~0.2 nmol/kg; Poggemann et al., 2017), which is also in good agreement with the benthic assemblage analyses from the same cores. Therefore, although it may be difficult to exclude the influence of NADW during the Holocene, the significant high intermediate $Cd_w$ during the late Holocene does not correspond to the increased contribution of NADW, suggesting that our initial interpretation could be also maintained during the Holocene.

b) Also, in line 329 (and thereafter), the authors, for the first time, mention NADW, claiming that this water mass would dominate during the early Holocene at site MD77-191. How does this claim compare to the modern water mass distribution in the area? Is it not true that most of the deep waters in the Indian Ocean are mixes of various end members, of which NADW is just one? The only place original (largely unmixed) NADW occurs is off the southeast coast of Africa, with the northward propagation blocked by the Davie Ridge (although there is some discussion in relation to a potential northward spillover occurring). In order to substantiate their argument, a) the hydrography section needs improvement and b) there needs to be a more in-depth explanation how (even contributions) of a deep water mass, currently occurring below ~2km in in the Mozambique channel, affect sediment cores at true intermediate water depth. The latter changes affect the discussion of the entire Holocene record.

Answer: We fully agree with the reviewer that the discussion about the NADW has to be improved. In the modern northern Indian Ocean, the Indian Deep Water (IDW) lies between 1500 and 3800 m. The IDW forms from the mixing between NADW and Circumpolar Deep Water (Talley et al., 2011). As already detailed in the answer to the upper comment, multiple geochemical proxies obtained from core MD77-191 (Arabian Sea) and MD77-176 (northern BoB) as well as previous studies (Yu et al., 2018; Ma et al., 2019; 2020) have provided strong evidence for the increased contribution of well-ventilated NADW during the Holocene. Thus, although the influence of NADW on the intermediate $Cd_w$ cannot be fully discarded during the Holocene, our records suggest the dominated role of surface productivity in controlling $Cd_w$ records during the Holocene, and thus, our conclusion will not change but will be better discussed (the modifications of NADW discussion are explained in Reply a)); indeed, in order to take into account these comments, we developed the discussion about NADW during the modern and Holocene periods (revised Section 2).

c) There is some inconsistency regarding the description (interpretation?) of the habitat of the various benthic foraminifera species used in the study. In lines 141 and 142, the authors state that *C. pachyderma* is an epifaunal species. In contrast, in 289 and 290 they state that it is a shallow infaunal species. This needs to be clarified and consistently used throughout the manuscript.

Answer: We have corrected this mistake in the revised manuscript to be:

*C. pachyderma* is a shallow infaunal species.

d) At times the description of results/findings is too generic. As an example in lines 364 and following, a number of comparisons are made regarding the similarity of records. Generally, on longer time scales, yes there is some similarity. It should be pointed out though that there are also substantial differences at the millennial scale. This is particularly relevant for the comparison between Corg and *H. elegans* Cdw records. This needs a better wording.

Answer: We agree that these sentences are not well-written because we only focused on the long time scale variations, so we used this comment to improve the following discussion of the comparison.

Despite a lower resolution for MD77-191 *H. elegans* $Cd_w$ records, when compared to the $C_{org}$ and the *G. bulloides* percentage from core SK237 GC04, all of them seem to exhibit similar trends at the long time scale from the last deglaciation to Holocene; however, some little discrepancies can be observed at millennial time scales, especially during the late Holocene (Fig. 5).

e) (minor point) Figure 6 needs a better embedding/explanation in the manuscript. Some of the records are neither explained in the main text nor in the figure caption.

Answer: Thanks for this reminding. We have detailed these records explanation in the main text as the following paragraph, especially for figures 6 d-f records.

These changes are consistent with the weakened the summer monsoon intensity, with less rainfall during the late Holocene, as observed in the BoB using core MD77-176 seawater $\delta^{18}O$ and core SO188-342KL $\delta D_{Alk-ic}$ records (Marzin et al., 2013; Contreras-Rosales et al., 2014; Figs. 6 e-f). In addition, this is also strongly supported by the $\delta^{13}C_{wax}$ records from the Lonar Lake over the Indian continent (Sarkar et al., 2015; Fig. 6d) and a progressive increase in monsoon summer winds to the South of India (Bassinot et al., 2011).

Besides, in the figure and caption, we also made the brief description.

Overall, there are some useful data in this manuscript. The discussion of the data and subsequent interpretation lacks maturity at this stage and requires improvement. A moderate to major revision is required.

---

## Author Comment (AC2) · 26 Jul 2021

**Reply to reviewer #2 comments:**

We thank Reviewer André Bahr for providing helpful comments on our manuscript, they have been carefully considered. Please find below our answers to these comments.

Review of Changes in productivity and intermediate circulation in the northern Indian Ocean since the last deglaciation: new insights from benthic foraminiferal Cd/Ca records and benthic assemblage analyses by Ma et al.

The authors present benthic foraminiferal assemblage records and Cd/Ca data from the western Arabian Sea and Bay of Bengal (BoB) to investigate surface primary productivity (PP) and intermediate water mass variability in the context of the last deglaciation and Holocene climatic evolution. The authors find that Cd/Ca is primarily controlled by PP during the Holocene which mirrors monsoonal intensity. Notably, a strong monsoon is inferred to suppress PP in both areas due to enhanced run-off which increases upper ocean stratification in the BoB and reduced Ekman-upwelling off India as a result of decreased wind stress. During the deglaciation, the authors infer a dominance of water mass changes in driving the Cd/Ca signal, showing an enhanced advection of AAIW into the northern Indian Ocean during YD and HS1.

In general, the data and its interpretation appear mostly sound and in line with existing concepts about the paleoceanography of the Arabian Sea and the BoB as well as the influence of the monsoon on the PP in these areas. In this respect I find it noteworthy that the data (i) supports the presumed E-W dipole between strong upwelling off Oman and weak upwelling off India, and (ii) that maximum monsoon induced run-off in the BoB apparently suppresses PP due to strong stratification, despite riverine nutrient input should enhance plankton blooms on surface level. The authors mention stratification as an explanation rather briefly, however, I would encourage the authors to devote one or two more sentences on this issue (see also my detailed comment).

Answer: The reviewer suggests that the arguments for the monsoon intensity influence on the PP in the southeastern Arabian Sea and northeastern BoB during the Holocene should be stronger. In order to reinforce the discussion, we developed this part in Section 5.3., improving the quality of the discussion about the contribution of nutrient inputs from rivers in these areas, even if this process seems not to be significant. We are now providing the following explanation in the manuscript:

However, the distribution of chlorophyll in surface water of the western BoB suggests a low annual productivity, indicating that the BoB is not significantly influenced by the riverine nutrient input (Zhou et al., 2020). Thus, it is likely that this increase in fresh water drove pronounced ocean stratification in the northeast BoB, which could impede the nutrient transfer from deep layer to the euphotic upper seawater column, and then inducing low productivity.

I can also follow the arguments for the inferred intrusion of AAIW into the northern Indian Ocean during HS1 and YD, which agrees with the well-documented enhanced northward protrusion of this water mass in the Atlantic Ocean. However, I am not convinced by the way the authors come to this conclusion, which is based on the claimed mismatch between increasing *G. bulloides* abundances and decreasing Cdw estimates during YD and HS1. As depicted in the figure below, both records essentially follow the same trend, also bearing in mind that the resolution of

Cdw is relatively low during YD and HS1. Hence, the proposed anti-correlation between *G. bulloides* abundances and Cdw seems to be an overstatement.

Irrespective of this problem, the good match of Cdw and $\delta^{13}C$ with AAIW reference records make it reasonable to assume that the Cdw values in deed capture water mass variability between HS1 and YD (Fig. 7). Hence, the interpretation at the end seems correct, but it is more likely that the relatively modest increase in PP during the YD/HS1 appear to have had a negligible influence of the Cdw. Only if PP is really high (such as in the mid-Holocene) Cdw is dominated by PP, as also indicated by the very high values > 1.0. The authors are somewhat over-confident regarding the use of Cd/Ca as a water mass tracer in the such potentially highly productive areas and might consider toning down their argumentation.

Answer: We fully agree with the reviewer that the conclusion - *H. elegans* $Cd_w$ displays an anti-correlated compared with the *G. bulloides* abundances during YD and HS1 - seems to be overstated because of the relatively low resolution during the last deglaciation. In order to take this comment into account, we slightly modified the discussion by separating the interpretation of the comparison between $Cd_w$ and *G. bulloides* abundances (revised Section 5.4). We are now providing the following corrections in the manuscript:

Significant decreases in *G. bulloides* relative abundance of cores SK237 GC04 (Naik et al., 2017) and MD77-191 records were observed from the HS1 to B-A (Bassinot et al., 2011), and thereafter slight increases occurred in the YD (Fig. 5). These high values at both core sites during the HS1 and YD may indicate an enhanced surface productivity during these intervals (Fig. 5). This should have led to increased intermediate $Cd_w$ and organic matter preservation under low oxygen concentration conditions during the HS1 and YD. However, despite a low resolution for the MD77-191 $Cd_w$ record during the last deglaciation, we do not observe high values of intermediate $Cd_w$ during the HS1 and YD ($\sim$0.6 nmol/kg) compared with the late Holocene (~1.59 nmol/kg), especially at 16.5-16 cal kyr BP. Although we cannot fully discard the influence of surface productivity on the intermediate $Cd_w$ in these time intervals, this apparent discrepancy seem to provide another evidence for the influence of changes in water masses and/or ventilation during the HS1 and YD, as already demonstrated by previous studies and proxies in the northern Indian Ocean (Bryan et al., 2010; Yu et al., 2018; Ma et al., 2019; 2020).

While the manuscript is well written, the Figures might benefit from rearrangement to make the discussion more easier to follow (cf. detailed comments below).

Given some moderate revisions I support publication of this study which represent an important contribution to our understanding of the deglacial evolution of the Indian Ocean.

[Figure]

Figure 5 with Cd/Ca from MD77-191 in blue, illustrating that Cd/Ca follows *G. bulloides*-abundances (even more in the more high-resolution data of adjacent core GC04).

Answer: Although the new figure suggested by the reviewer allows directly comparing the Cd/Ca with the % *G. bulloides* record, it may be difficult to decipher both records. Moreover, the reviewer superimposed records from two different cores, even if the global trends are very close to each other. Thus, we prefer to keep the Fig. 5 in its original version.

**Detailed comments**

Line 78: The motivation for the study is rather weak (essentially: we know little about the paleoproductivity of the BoB). It would be good to more explicitly state why we should care about this issue.

Answer: We agree with the reviewer and add some sentences to clarify the motivation as is shown in the following paragraph.

   By contrast, little is known about the paleoproductivity of the BoB, especially its links to changes in monsoon precipitation (Phillips et al., 2014; Zhou et al., 2020). Consequently, studying paleoproductivity in the northeastern Indian Ocean will also allow us to complete understand the influence of monsoon climate changes in tropical ocean ecology at different timescales. Besides, as the benthic foraminiferal Cd/Ca is a promising proxy to reconstruct the intermediate-deep water nutrient content (e.g., Boyle and Keigwin, 1982; Tachikawa and Elderfield, 2002; Came et al., 2008; Poggemann et al., 2017; Valley et al., 2017), most of the studies referred to above have reconstructed deep-intermediate water masses in the past (e.g., Came et al., 2008; Bryan and Marchitto, 2010; Poggemann et al., 2017; Valley et al., 2017), only few works indicate the relationship between the intermediate water masses nutrient and surface productivity (Bostock et al., 2010; Olsen et al., 2016).

L. 85: "estimate past changes in the nutrient content, since the last deglaciation, over the last 17 kyr BP." The last part is redundant.

Answer: We have corrected this sentence in the revised manuscript to be:

"estimate past changes in the nutrient content since the last deglaciation."

l. 100: "of the planktonic…"

Answer: It has been done.

l. 101 (and elsewhere): avoid using "." as multiplicator

Answer: It has been done.

L. 109: "Arabian High Salinity Waters" (all capitals)

Answer: Corrected.

L. 118: Neither Fig. 1 nor S1 show salinity.

Answer: We fully agree this comment and removed it in the revised manuscript.

L. 131: "northern intermediate …" (no capitals)

Answer: Corrected.

Section 3: Please also include the statistical methods used in the study in the Methods chapter. Which program did you use to perform the PCA? Did you use a correlation or variance/covariance matrix?

Answer: We have corrected this sentence in the revised manuscript to be:

In order to describe major faunal variations, we performed principal component analysis (PCA) on the variance-covariance matrix using the PAST software (Version 3.0, Hammer et al., 2001).

Section 3.1.: Regarding the design of the study, I wonder why the authors decide to use four different species, when *H. elegans* is available as a well-documented, faithful recorder of bottom water Cd/Ca. What was the rationale to use the three calcitic species, especially as they include infaunal dwellers which are naturally not the best suited for detecting bottom water fluctuations?

Answer: We fully agree with the reviewer that *H. elegans* is a well-documented and faithful recorder of bottom water Cd/Ca. Indeed, Tachikawa and Elderfield (2002) indicated that due to the lower partition coefficients, the infaunal benthic foraminifera could record Cd/Ca values similar to *Cibicidoides*, despite elevated pore water Cd. Thus, many previous studies used both *Cibicidoides* and *Uvigerina* in paleoceanographic reconstructions (e.g., Marchitto and Broecker, 2006; Makou et al., 2010; Umling et al., 2018; 2019). At core MD77-191 site, we could provide continually calcite benthic species samples with different microhabitat (*Cibicidoides pachyderma*, *Uvigerina peregrina*, and *Globobulimina* spp.). Therefore, we performed Cd/Ca analyses on these benthic species to improve understanding of possible species level differences and microhabitat effects on the benthic Cd/Ca records. We clarified this point in the revised manuscript as:

In order to improve understanding of possible species level differences and microhabitat effects on the benthic Cd/Ca records, we analyzed Cd/Ca in three calcite (*Cibicidoides pachyderma*, *Uvigerina peregrina*, and *Globobulimina* spp.) and one aragonite (*Hoeglundina elegans*) benthic

foraminiferal species from core MD77-191.

L. 204-205: you might omit "over the last deglaciation"; add an "a" before "significant decrease"
Answer: We have corrected this sentence in the revised manuscript.

L. 228 etc., regarding the PCA results: You show 2 PCs which explain 61% of the total variance. What's about the other PCs, how much variance to they explain and what was the rationale to limit the investigations to those two PCs?
Answer: The total variance of other PCs is 49% which is shown in the following figure. Compared with the total variance of PC1 (42%) and PC2 (19%), PC3 is the largest one and only explains 8% of the total variance for the rest PCs. The species composition consist of *Hoeglundina elegans* (0.66), *Globobulimina* spp. (0.22) (Positive loadings), *Uvigerina peregrine* (-0.59), *Cibicidoides pachyderma* (-0.21) (Negative loadings). It seems that the main composition of assemblages (PC3) is quite similar to PC1, and does not show more information about the bottom conditions. Thus, we only use PC1 and PC2 in the manuscript to recognize the three assemblages.

[Figure]

Figure: the variance of total PCs for core MD77-191

When the authors discuss the loadings of the individual PCs, they assign particular species with a very low loading to individual assemblages. PC1 for example is very much dominated by *B. aculeata* (+0.84 loading); the denoted loadings of -0.07 and less for *B. manginata*, *C. wuellerstorfi*, *G. subglobosa* (Table 1) appear to be rather insignificant. The same applies for PC2 which has high loadings of +0.42 and -0.62 for *S. bulloides* and *H. elegans*, respectively; I doubt that e.g. *G. soldanii* with a loading of 0.07 has a significant relevance to PC 2. Please reconsider the discussion of the PC 1 and PC 2 accordingly. You might also consider providing a bi-plot for PC 1 and PC 2 as an extra figure.
Answer: We agree that the dominant species (*B. aculeata*, *S. bulloides* and *H. elegans*) make a significant contribution for these three assemblages, respectively. We recognized three benthic assemblages based on the positive and negative loadings of different PCs. Despite the loading values of *B. manginata*, *C. wuellerstorfi* and *G. subglobosa* are much less compared with the dominant species (*B. aculeata, S. bulloides and H. elegans*), both these lower loadings species are environmental sensitive species, associated with different bottom water conditions (e.g., Corliss et al., 1986; Schmiedl et al., 1998; Almogi-Labin et al., 2000). Thus, it seems reliable to use these benthic spices for the interpretation of assemblages. In addition, we have plot the PC1 and PC2 records together in Figure 3, so we prefer not to add an extra figure about the bi-plot for PC 1 and PC 2 in the manuscript.

L. 249: "aragonite": change into "argonitic"

Answer: It has been done.

L. 262: The paragraphs discussing Cdw repeat in large parts what have been written about the Cd/Ca ratio in the Results chapter. Please avoid such duplication. The same also applies for Figures 2 and 4. Figure 2 might be moved into the supplement.

Answer: We agree with the reviewer that the description of $Cd_w$ records is similar with the Cd/Ca ratios, we modified and remove some sentences in the section 5.1 about the discussion of intermediate water $Cd_w$ results from the Northern Indian Ocean to avoid the duplication. We are now providing the following descriptions in the manuscript:

The intermediate $Cd_w$ results based on the *H. elegans* Cd/Ca values of core MD77-191, range from 0.5 to 3.1 nmol/kg since 17 cal kyr BP (Fig. 4a), with a core top value of 0.80 nmol/kg in agreement with the estimated intermediate water depth modern $Cd_w$ (~0.83 nmol/kg) in the northern Indian Ocean (Boyle et al., 1995). The intermediate $Cd_w$ was also calculated from calcite benthic species *C. pachyderma*, *U. peregrina* and *Globobulimina* spp. from core MD77-191, with values ranging between 0.53-1.48 µmol/mol, 0.52-1.04 µmol/mol and 0.26-0.65 µmol/mol, respectively (Fig. 4a). The $Cd_w$ values of *C. pachyderma* and *U. peregrina* are within the same range. However, the *H. elegans* $Cd_w$ values are higher than those from the two calcite species, especially during the Late Holocene. Moreover, the core top data of *C. pachyderma* and *U. peregrina* are also lower (~ 0.7 and 0.69 nmol/kg, respectively) than the modern estimated $Cd_w$ data (~ 0.83 nmol/kg) in the northern Indian Ocean (Boyle et al., 1995) (Fig. 4a). These depleted $Cd_w$ values may be related to the benthic foraminiferal microhabitat effect; indeed, *U. peregrina* is known to be strictly a shallow infaunal species, as well as *C. pachyderma* (Fontanier et al., 2002), differing from strictly epifaunal taxa, such as *Cibicidoides wuellerstorfi* (Mackensen et al., 1993).

Besides, the deep infaunal *Globobulimina* spp. $Cd_w$ displays relatively much lower values and does not exhibit strong variations compared to the other species investigated in this study, displaying a general increasing trend from the last deglaciation to the Holocene. As *Globobulimina* spp. correspond to deep benthic infaunal species, this result may indicate a stable nutrient content of pore water, as compared to other benthic taxa associated with bottom water (Fig. 4a). Thus, when tracking past changes in the bottom water $Cd_w$ concentrations, the use of a strictly epifaunal species living at the water-sediment interface such as *H. elegans* appears to be more robust than using endofaunal species that live in contact with pore water.

Relative variations in the $Cd_w$ obtained from *C. pachyderma* and *U. peregrina* are in good agreement with the records obtained on *H. elegans*. Variations of *H. elegans* $Cd_w$ during the last deglaciation indicate a decrease of about ~0.6 nmol/kg in the HS1 and YD periods, with a slight increase (0.9 nmol/kg) during the warm B-A. $Cd_w$ results from core MD77-191 indicate a shift from the last deglaciation (~0.7 nmol/kg) to the late Holocene (~1.59 nmol/kg). During the Holocene, the $Cd_w$ records display relatively low values of around 0.9 nmol/kg in the 10-6 cal kyr BP time interval, and show a major shift at around 6.4 cal kyr BP with values rising up to 3.1 nmol/kg.

The reviewer also suggests that Figure 2 might be moved to the supplementary information, but we do not fully agree with this comment. Indeed, comparing the *Globigerinoides ruber* $\delta^{18}O$ records of cores MD77-191and MD77-176 with GISP2 Greenland ice core $\delta^{18}O$ signal (Stuiver

and Grootes, 2000) in Figure 2 allows defining the key time intervals we focus, i.e. the Heinrich Stadial 1, the Younger Dryas, the Bølling-Allerød events and the EHCO. We think that it helps to clearly describe the results (Cd/Ca and benthic assemblages), as well as the following discussion. There, we prefer to use Figure 2 here rather than move it to the supplement.

L. 264: please add the modern Cdw to figure 4 for reference.
Answer: We improved figure 4 with this common. Please see new Figure 4.

[Figure]

**Fig. 4.** (a) Cd$_w$ records calculated based on the Cd/Ca of benthic foraminifera *Hoeglundina elegans* (black), *Cibicidoides pachyderma* (green), *Uvigerina peregrina* (blue), and *Globobulimina* spp. (orange) obtained from core MD77-191, (b) Cd$_w$ record from core MD77-176 reconstructed using *H. elegans* Cd/Ca, the red line is the smoothed curves using a five-point average. The red stars represent the modern Cd$_w$ (~0.83 nmol/kg) in the northern Indian Ocean (Boyle et al., 1995). The color shaded intervals and abbreviations are the same as in Figure 2.

Figures 2 and 4: The y-axes should have a common scale to enable direct comparison of the individual records. As shown, the Cd/Ca and Cdw records of *Globobulimina* spp. appear to show large fluctuations, however, compared to the other species these fluctuations are of rather minor importance (as stated in the text).
Answer: We provide new figures with a common scale for y-axes. Please see new Figure 2. The new Figure 4 is already shown in the reply for the above question "L. 264".

[Figure]

**Fig. 2.** (a) GISP2 Greenland ice core $\delta^{18}$O signal (Stuiver and Grootes, 2000). (b)-(c) *Globigerinoides ruber* $\delta^{18}$O records of cores MD77-191and MD77-176, respectively (Marzin et al., 2013; Ma et al., 2020). (d) Cd/Ca records of the benthic foraminifera *Hoeglundina elegans* (black), *Cibicidoides pachyderma* (green), *Uvigerina peregrina* (blue), and *Globobulimina* spp. (orange) obtained from core MD77-191; (e) Cd/Ca records of the benthic foraminifera *H. elegans* from core MD77-176. EHCO for Early Holocene Climate Optimum, YD for Younger Dryas, B-A for Bølling-Allerød and HS1 for Heinrich stadial 1.

L. 286: the core tops values mentioned here should be shown in the respective Figure for reference.

Answer: Corrected. Please see new Figure 4 in the reply for question "L. 264: please add the modern Cdw to figure 4 for reference".

L. 300-303: The final statement that *H. elegans* provides the most reliable Cd/Ca (or Cdw) data is not really surprising. As stated in the earlier comment I would appreciate if the authors could provide more arguments what they wanted to test/proof with including the other three species.

Answer: Please refer to the reply for question "Section 3.1.: Regarding the design of the study, I wonder why the authors decide to use four different species, when *H. elegans* is available as a well-documented, faithful recorder of bottom water Cd/Ca. What was the rationale to use the three calcitic species, especially as they include infaunal dwellers which are naturally not the best suited for detecting bottom water fluctuations?"

Indeed, we indicated in the manuscript in lines 270-277, which discussed deep infaunal

*Globobulimina* spp. $Cd_w$ results. Lines 286-293, we compared $Cd_w$ obtained from *C. pachyderma* and *U. peregrina* with records obtained on *H. elegans*, the depleted $Cd_w$ values of calcite infaunal species may be related to the microhabitat effect, thus *H. elegans* appears to be more robust than using endofaunal species that live in contact with pore water. Although we modified the sentences and structure of the discussion about intermediate water $Cd_w$ results in the section 5.1 to avoid the duplication, we still keep the comparison of these four benthic species $Cd_w$. Please refer to reply for question "L. 262: The paragraphs discussing $Cd_w$ repeat in large parts what have been written about the Cd/Ca ratio in the Results chapter. Please avoid such duplication. The same also applies for Figures 2 and 4. Figure 2 might be moved into the supplement".

L. 307 etc.: I think the presentation of the Assemblage data could be improved: 1) as stated earlier the presentation of the PCA results is not totally convincing;

Answer: As mentioned before (please refer to question Line 228), the total variance of PC1 (42%) and PC2 (19%) could explain 61% of the total variance, and for other PCs, the total variations are much less. Besides, the main composition of the rest PCs negative or positive loadings is dominated by the same benthic species as these recognized three assemblages (such as based on PC3, 8% total variation, the largest one among the rest PCs), it is difficult to glean more additional information from this regarding bottom conditions. Thus, it seems reliable that we recognize three assemblages in this paper.

2) Why do the authors start with Assemblage 3 not in the numerical order?

Answer: We interpreted our records in time order from the last deglaciation to Holocene.

3) If they use the Assemblages as environmental indicator they could plot the abundance of the respective Assemblages instead of individual foraminifera species in the results figure; they could also assign specific environmental parameters to each Assemblage in the Figures (e.g. Assemblage 1 = more/less productivity or oxygenation), which would help to more concisely convey the message of the study.

Answer: The two first-ranked principal components are selected to obtain species associations with demands to specific environmental conditions. Based on the positive and negative loadings of PC1, we recognized benthic assemblages 1 and 2, respectively. In addition, assemblage 3 was identified by the positive loadings of PC2. And then we explored to discuss the bottom water environmental condition changes, combining different major individual benthic taxa of the assemblages, thus, the abundance of respective assemblages may be not suitable as the environmental indicator. Therefore, we prefer to keep the figures in their original version.

L. 351: better write "Assemblage 1" instead of "fauna 1"
Answer: Corrected.

L. 353. "depleted *Globigerina bulloides* abundances" – replace by "low *G. bulloides* abundances"
Answer: It has been done.

L. 370: please refer more often to the respective Figures.
Answer: We have added the reference figures 3 and S2 in the revised version.

L. 398: a decreasing Cdw trend between 5.2 and 2.4 cal kyr BP is not evident for me, the values are constantly very high during this time period.

Answer: We agree with the reviewer that the $Cd_w$ values are constantly high during 5.2-2.4 cal kyr BP, especially compared with the early Holocene. Thus we have corrected this description in the revised version as:

"reach a maximum during the late Holocene".

L. 405-406: the relation of stratification and PP should be discussed in more detail (see my general remark).

Answer: We used this comment to improve the discussion of the manuscript. We are now providing the following explanation in the manuscript:

However, the distribution of chlorophyll in surface water of the western BoB suggests a low annual productivity, indicating that the BoB is not significantly influenced by the riverine nutrient input (Zhou et al., 2020). Thus, it is likely that this increase in fresh water drove pronounced ocean stratification in the northeast BoB, which could impede the nutrient transfer from deep layer to the euphotic upper seawater column, and then inducing low productivity.

L. 431-432: The authors refer to summer insolation – pleas show it in an appropriate figure (e.g. Fig. 5)

Answer: Bassinot et al. (2011) indicated that the ITCZ location shifted northward when boreal summer insolation reached a maximum in the early Holocene, associated with the enhanced summer monsoon wind intensity and a decrease in the Eckman pumping in the southern tip of India. This configuration then induced a decrease in surface productivity in the southeastern Arabian Sea. We added the summer insolation variations in the modified Figure 6, please see the new Figure 6.

[Figure]

**Fig. 6.** (a) the solar insolation at 10 °N in summer (Laskar et al., 2004). (b) and (c) intermediate $Cd_w$ calculated from *H. elegans* obtained from MD77-176 and MD77-191, respectively. (d) Lonar Lake $\delta^{13}C_{wax}$ record (Sarkar et al., 2015). (e) $\delta D_{Alk-ic}$ record from core SO188-342KL (Contreras-Rosales et al., 2014). (f) Seawater $\delta^{18}O$ anomaly obtained from MD77-176 (Marzin et al., 2013). The color shaded intervals and abbreviations are the same as in Figure 2.

L. 453: The reference to Figure 7: should't it be rather Fig. 6?

Answer: Corrected.

L. 457-458: "Thus, we do not expect that surface productivity played an important role during the last deglaciation." This statement is odd, as it has been discussed at great length that PP is influencing Cd/Ca. The following "In addition,…" does also not fit as the following sentence does not support the above notion of PP playing an unimportant role.

Answer: We agree with the reviewer that these sentences seem to be odd here and removed them in the revised version.

L. 462 etc: I am not convinced by the statement that increasing *G. bulloides* abundances during HS1 and YD are in conflict with both, Corg and Cdw records. With regards to Corg I agree that it declines opposite to the trend in *G. bulloides*, however, Corg does not only depend on PP but also on preservation, and potentially sedimentation rate (one way to check the influence of sediment accumulation would be to compute Corg accumulation rates). However, Cdw rather follows *G. bulloides* abundances, at least it is not anti-correlated, as one might infer from the text.

Answer: We agree with the reviewer and correct these sentences in the revised version. We also fully agree with the reviewer that the $C_{org}$ could depend on PP and/or preservation, this has been also indicated in the Manuscript (lines 358-360). In this study, we mainly focus on the *G. bulloides* abundance to reflect the paleoproductivity in this region. The modifications of revised lines 462 etc are explained in Reply the second general comment).

L. 482-487: the sentence is too long and complicated, please rephrase.

Answer: We have corrected this sentence in the revised manuscript to be:

   Thus, as the benthic $\delta^{13}C$ values collected from the north Indian Ocean could better constrain the influence of AAIW in the two studied cores (Naqvi et al., 1994; Jung et al., 2009; Ma et al, 2019; 2020), we can also compare the range values of AAIW $Cd_w$ from both studied cores with data from Atlantic and Pacific Oceans at intermediate water depth during the HS1 and YD ($Cd_w$, 0.3-0.9 nmol/kg; Umling et al., 2018; Valley et al., 2017).

L. 496: "increase" instead of "icrease"

Answer: Corrected.

L. 504: "the entire biological factory was related to reduced monsoon intensity" – see my general comment: this statement need more justification. The presence/absence of *G. bulloides* might well be influenced by stratification and surface water freshening, but does this apply to other primary producers as well?

Answer: Unfortunately, we have no other productivity indicators records available from core

MD77-191 (Arabian Sea).

**General remarks to the figures:**

It helps the reader if the authors state next to the core name where the core is located (e.g. within the BoB)

Answer: For figure 1, we have put the "Arabian Sea" and "BoB" names in the right place. And for these two figures, there are more information should be shown clearly, such as the general surface circulation and primary productivity distribution of the northern Indian Ocean. If we move the "Arabian Sea" and "BoB" names next to the core name, these two figures seems to be "confused", so we prefer to keep the core name in its original version.

Fig. 5: you might add here the benthic $\delta^{13}$C records of MD77-191 and -176 used to discussed water mass variability (cf. Fig. S3)

Answer: We do not fully agree with this comment since for Fig. 5, in order to examine the relationship between primary productivity and intermediate $Cd_w$ records of the studied cores, we compare MD77-191 $Cd_w$ results with different records associated to surface productivity from the same area (southeastern Arabian Sea). The similarity of the benthic $\delta^{13}$C-increases reflects the northward expansion of AAIW during the last deglaciation in the north Indian Ocean and Pacific Ocean (Pahnke and Zahn, 2005; Jung et al., 2009; Ma et al., 2019, 2020), the detailed interpretation based on the benthic $\delta^{13}$C records (including cores MD77-191 and MD77-176) have been discussed in these previous papers, thus we prefer to use Fig. S3 as the supplementary information to interpret the changes of intermediate water masses.

S1: please add the variable + unit to the color shading on the right side of the maps.
Answer: Corrected. Please see new Figure S1.

[Figure]

**Fig. S1.** a) – b) The Net primary productivity distribution in the Northern Indian Ocean during January and July, respectively. Maps based on MODIS chlorophyll-a, SST, PAR satellite data,

using the standard vertically Generalized Production Model (VGPM) (Behrenfeld and Falkowski, 1997) as the standard algorithm.

---

## Referee Report (RR1)

**Review of Ma et al. (revised version)**

In my view the authors did a great job to address the concerns raised by both Reviewers. In particular they now more clearly discuss the uncertainties of their $Cd_w$ record an the potentially varying influence of primary surface productivity and intermediate water properties. Hence, I suggest acceptance of the manuscript pending very minor revisions, which basically regard typos and ambiguous wording.

In particular the abstract would benefit from more explicitly writing which factors are driving the $Cd_w$ record during which time period. The authors all state this is the abstract but it reads rather indirect. Below a recommendation for rephrasing (lines 22-29), please feel free to adopt or dismiss these suggestion.

"These results suggest that during the last deglaciation Cdw variability was primarily driven changes in intermediate water properties, indicating an enhanced ventilation of intermediate-bottom water masses during both Heinrich Stadial 1 and Younger Dryas (HS1 and YD, respectively). During the Holocene, however, surface primary productivity appeared to have influenced Cdw more than intermediate water mass properties. This is evident during the early Holocene (from 10 to 6 cal kyr BP) when benthic foraminiferal assemblages indicate that surface primary productivity was low, resulting in low intermediate water Cdw at both sites. Then, from ~ 5.2 to 2.4 cal kyr BP, surface productivity increased markedly, causing a significant increase in the intermediate water Cdw in the southeastern Arabian Sea and the northeastern BoB. "

L. 48: "contribute to up to..:"

L. 86: "only few works indicate the" – do you mean "investigate"?

L. 114: avoid using "." as multiplier

L. 142: "linked to increased primary productivity"

L. 148: what do you mean by "species level differences"? Inter-species offsets?

L. 335: "which is characterized by the well-ventilated and depleted nutrient" – something is missing here

L. 340: "Benthic foraminifera" - no capital

L 357: "in the bottom water"

L. 359: "Assemblage 1" – no capital

L. 361: *Globigerina* can be abbreviated

L. 375: "little discrepancies" – better use "small-scale" instead of little. However, I think the discrepancies are not so small, although I agree that the long-term trend is similar.

L. 423: "from the deep layer" – please also specify which deep layer you mean. Intermediate waters? Thermocline waters?

L. 480-481: "another evidence for the influence of changes in water masses and/or ventilation during the HS1 and YD, as already demonstrated by" – this sentence might be rephrased as it undersells the results; especially "another evidence" and "as already" sounds like the data adds nothing new to the

existing records which is not the case). I would suggest to write "…  evidence for the influence of changes in water masses and/or ventilation during the HS1 and YD, in line with…"

L. 498-499: "enhanced northward flow of southern sourced intermediate water mass AAIW" – there is something missing here

---

## Author Response (AR3)

**Author response: version 1**

**Ruifang Ma**
Université Paris-Saclay, Laboratoire GEOPS, UMR 8148
Campus scientifique d'Orsay, Bâtiment 504
91405 Orsay, France
maruifang89@hotmail.com

November 17$^{th}$, 2021

Dear editor,

We have submitted a new version of manuscript online, taking into account most of the comments made during the review process. First, we would like to thank you for your help in each step of the submission and review processes, as well as the constructive comments provided by you and the two reviewers. The main points relied on i) the interpretation of the geochemical proxies, related to changes in the circulation and/or productivity, that should be more critical and ii) a better assessment of the water masses at studied sites (and, especially, the occurrence of North Atlantic Deep Water NADW). We reinforced these two points in the new version of the manuscript, following the strategy suggested by both reviewers.

In details, we answered to the main comments about the uncertainties on the interpretation of the $Cd_w$ values, especially during the last deglaciation. We improved this point by developing the discussion about the $Cd_w$ as a proxy for the productivity changes and changes in the circulation (see details in the following answer to the reviewer as well as the new paragraph in the revised manuscript in section 5.4, lines 500-517). To better assess the contribution of mixed various end members for the deep water in Indian Ocean, we developed the discussion about deep water masses in modern period (section 2, lines 123-125) and also the contribution of NADW at the studied sites during the Holocene in the revised manuscript (section 5.2, lines 359-365 and 405-409). As previous studies based on multiple geochemical proxies (benthic $\delta^{13}C$ values, B-P age offsets, $\varepsilon_{Nd}$, $\delta^{18}O_{ivc}$ and Mg/Li) obtained from same cores and/or near core sites suggest an increased influence of NADW during the Holocene (Ahmad et al., 2008; Raza et al., 2014; Yu et al., 2018; Naik et al., 2019; Ma et al., 2019; 2020), we still believe in our original interpretation, but we reinforced the discussion, as developed in the following detailed answers.

The second reviewer, Dr. André Bahr, also underlined some confusion about the way to interpret elemental ratios during the last deglaciation. To correct that, we developed the interpretation of the comparison between $Cd_w$ and *G. bulloides* abundances (section 5.4, lines 500-517), to reinforce the discussion about the influence of changes in water masses and/or ventilation at the studied site during the last deglaciation. Moreover, most of the other comments suggested by Dr. André Bahr were taken into account into the final version of the manuscript, as detailed below.

To finish, we decided to add the anonymous reviewer#1 as well as André Bahr in the 'Acknowledgements' part for their help to improve the quality of the manuscript.

We hope that the new version of the manuscript meets the high-quality standards set for *Climate of the Past* publications.

In the following, we provide detailed answers for the specific reviewer comments.

**Reply to reviewer #1 comments:**

We thank Reviewer 1 for providing helpful comments on our manuscript, they have been carefully considered. Please find below our answers to these comments.

Review of manuscript "Changes in productivity and intermediate circulation in the northern Indian Ocean since the last deglaciation: new insights from benthic foraminiferal Cd/Ca records and benthic assemblage analyses" by Ma et al.. This manuscript presents data from sediment cores from the northern Indian Ocean (Arabian Sea and Bay of Bengal), comprising geochemical time series generated on benthic foraminifera as well as census data for planktic and benthic foraminifera. Based on these data, the authors deduce changes in monsoon driven changes in productivity mainly dominating the various records during the Holocene and changes in intermediate water chemistry during the deglaciation. In principle the authors present interesting data and some of the interpretations appear justified. There are, however, a number minor and more major issues, preventing recommending publication as is at this stage. These are some of the issues:

a) The biggest issue is related to the lack of constituency of interpreting the Cdw records. In lines 450-453 the authors claim that the Cdw values during the deglaciation are lower than during the Holocene. First, this statement is only correct if longer term averages are considered. On short time scales (which need to be considered, given that this is a chapter on millennial scale change),

the youngest Cdw data in core MD77-191 (2-1.5 Ka BP) are comparable to YD and HS1 values. Up to this point a big effort has gone into establishing Cdw as reflecting productivity variations at the sea surface and the related flux of organic carbon. Now the focus shifts to bottom water ventilation changes being recorded. If general water ventilation would play are role in setting the recorded Cdw values, this has to apply to the Holocene too and would therefore need to be considered there too. Interestingly, the authors do involve water ventilation during the Holocene in relation to the carbon isotope and census data, but not very much in relation to the Cdw records. Also, if the general interpretation for the Holocene section is used, why is there no change in the Cdw record around 16-16.5 Ka BP? During this time, high *G. bulloides* concentrations (highest in the entire MD77-191 record) in the same core are shown in figure 5. High concentrations of *G. bulloides* strongly support the notion of enhanced productivity, as the authors themselves assume in case of the Holocene changes *G. bulloides* concentrations. Around 16-16.5 Ka BP the high *G. bulloides* concentrations are not reflected in the Cdw data. This would suggest that the Cdw water are not very reflective of surface productivity changes, casting doubts on parts of the Holocene storyline. This would need to be addressed in a revised version, not only in this section but in large parts of the manuscript.

Answer: The two main comments made by the first reviewer concerns i) the influence of surface productivity changes on the intermediate $Cd_w$ records at millennial time scale during the last deglaciation (16-16.5 cal kyr BP) and the Holocene (2-1.5 cal kyr BP) and ii) the contribution of intermediate water circulation variations, especially during the Holocene. We fully agree with the reviewer that there may be some questions interpreting the $Cd_w$ records, especially at the millennial scale. Changes in intermediate $Cd_w$ values of benthic foraminifera can be influenced by different processes such as surface productivity, changes of the water mass sources and/or ventilation (e.g., Came et al., 2008; Bostock et al., 2010; Olsen et al., 2016; Poggemann et al., 2017; Yu et al., 2019). First, at the Arabian Sea site, we suggest that the observed significant increase of intermediate $Cd_w$ values from the last deglaciation (~0.7 nmol/kg) to the late Holocene (~1.59 nmol/kg) could be associated with the surface productivity; indeed, this is supported by the *G. bulloides* record from the same core and another one close to the studied site (Bassinot et al., 2011; Naik et al., 2017), as well as by previous studies suggesting that increased $Cd_w$ values (>1 nmol/kg) could correspond to elevated surface productivity (Bostock et al., 2010; Olsen et al., 2016).

However, we also agree with the reviewer that there is some mismatch between the increased *G. bulloides* abundances and the decreased $Cd_w$ values obtained from the same core MD77-191 during the last deglaciation, especially at around 16-16.5 cal kyr BP. As the resolution of both records from core MD77-191 is relatively low during YD and HS1, it seems more reasonable to use the high-resolution *G. bulloides* abundances records from the near core site SK237 GC04 (1245m, southeastern Arabian Sea, Naik et al., 2017), reflecting the surface productivity changes in this area. Therefore, we observe that although the *G. bulloides* abundances from the Arabian Sea display an increasing trend during HS1 and YD events, these modest increases in surface productivity are synchronous with low $Cd_w$ values (~0.6 nmol/kg) during HS1 and YD, much lower compared to the $Cd_w$ results during the late Holocene (~1.59 nmol/kg). Thus, we suggest that the variations of surface productivity could not be the main control at this studied site during the last deglaciation, and thus we proposed that changes in the circulation can explain the observed results. Indeed, the influence of intermediate water masses variations has also been

demonstrated by many proxies ($\varepsilon_{Nd}$, benthic $^{13}$C, B-P age offsets and $CO_3^{2-}$) in the northern Indian Ocean (Bryan et al., 2010; Yu et al., 2018; Ma et al., 2019; 2020). These previous studies have been summarized and detailed discussed in the manuscript (lines 465-507). Consequently, the conclusion that the influence of changes in water masses and/or ventilation on $Cd_w$ records from the northern Indian Ocean during the HS1 and YD may be still reasonable. In order to take this comment into account, we modified the discussion in section 5.4 (lines 500-517) as below:

Significant decreases in *G. bulloides* relative abundance of cores SK237 GC04 (Naik et al., 2017) and MD77-191 records were observed from the HS1 to B-A (Bassinot et al., 2011), and thereafter slight increases occurred in the YD (Fig. 5). These high values at both core sites during the HS1 and YD may indicate an enhanced surface productivity during these intervals (Fig. 5). This should have led to increased intermediate $Cd_w$ and organic matter preservation under low oxygen concentration conditions during the HS1 and YD. However, despite a low resolution for the MD77-191 $Cd_w$ record during the last deglaciation, we do not observe high values of intermediate $Cd_w$ during the HS1 and YD ($\sim$0.6 nmol/kg) compared with the late Holocene (~1.59 nmol/kg), especially at 16.5-16 cal kyr BP. Although we cannot fully discard the influence of surface productivity on the intermediate $Cd_w$ in these time intervals, this apparent discrepancy seems to provide another evidence for the influence of changes in water masses and/or ventilation during the HS1 and YD, as already demonstrated by previous studies and proxies in the northern Indian Ocean (Bryan et al., 2010; Yu et al., 2018; Ma et al., 2019; 2020).

We also would like to thank the reviewer to point out that the influence of water mass ventilation on the intermediate $Cd_w$ during the Holocene should be better discussed. In order to clarify this point, we added some new parts in section 5.2 (lines 359-365, 405-409) about the possible contribution of past water masses changes to the $Cd_w$ records obtained from northern Indian Ocean during the Holocene. Briefly, increased benthic $\delta^{13}$C values and B-P age offsets, as well as depleted $\varepsilon_{Nd}$, $\delta^{18}O_{ivc}$ and Mg/Li results obtained from MD77-176 and MD77-191, suggest the increased influence of NADW (Yu et al., 2018; Ma et al., 2019; 2020); indeed, deep-water masses can contribute to intermediate water masses in the Northern Indian Ocean by upwelling when flowing northward (Talley et al., 2011; Naqvi et al., 1994). NADW is characterized by the fresh, well-ventilated and depleted nutrient (modern $Cd_w$, $\sim$0.2 nmol/kg; Poggemann et al., 2017), which is also in good agreement with the benthic assemblage analyses from the same cores. Therefore, although it may be difficult to exclude the influence of NADW during the Holocene, the significant high intermediate $Cd_w$ during the late Holocene does not correspond to the increased contribution of NADW, suggesting that our initial interpretation could be also maintained during the Holocene.

b) Also, in line 329 (and thereafter), the authors, for the first time, mention NADW, claiming that this water mass would dominate during the early Holocene at site MD77-191. How does this claim compare to the modern water mass distribution in the area? Is it not true that most of the deep waters in the Indian Ocean are mixes of various end members, of which NADW is just one? The only place original (largely unmixed) NADW occurs is off the southeast coast of Africa, with the northward propagation blocked by the Davie Ridge (although there is some discussion in relation to a potential northward spillover occurring). In order to substantiate their argument, a) the hydrography section needs improvement and b) there needs to be a more in-depth explanation how (even contributions) of a deep water mass, currently occurring below ~2km in in the Mozambique

channel, affect sediment cores at true intermediate water depth. The latter changes affect the discussion of the entire Holocene record.

Answer: We fully agree with the reviewer that the discussion about the NADW has to be improved. In the modern northern Indian Ocean, the Indian Deep Water (IDW) lies between 1500 and 3800 m. The IDW forms from the mixing between NADW and Circumpolar Deep Water (Talley et al., 2011). As already detailed in the answer to the upper comment, multiple geochemical proxies obtained from core MD77-191 (Arabian Sea) and MD77-176 (northern BoB) as well as previous studies (Yu et al., 2018; Ma et al., 2019 and 2020) have provided strong evidence for the increased contribution of well-ventilated NADW during the Holocene. Thus, although the influence of NADW on the intermediate $Cd_w$ cannot be fully discarded during the Holocene, our records suggest the dominated role of surface productivity in controlling $Cd_w$ records during the Holocene, and thus, our conclusion will not change but will be better discussed (the modifications of NADW discussion are explained in Reply a)); indeed, in order to take into account these comments, we developed the discussion about NADW during the modern and Holocene periods (section 2, lines 123-125).

c) There is some inconsistency regarding the description (interpretation?) of the habitat of the various benthic foraminifera species used in the study. In lines 141 and 142, the authors state that C. pachyderma is an epifaunal species. In contrast, in 289 and 290 they state that it is a shallow infaunal species. This needs to be clarified and consistently used throughout the manuscript.

Answer: We have corrected this mistake in the revised manuscript (line 152) to be:

   *C. pachyderma* is a shallow infaunal species.

d) At times the description of results/findings is too generic. As an example in lines 364 and following, a number of comparisons are made regarding the similarity of records. Generally, on longer time scales, yes there is some similarity. It should be pointed out though that there are also substantial differences at the millennial scale. This is particularly relevant for the comparison between Corg and H. elegans Cdw records. This needs a better wording.

Answer: We agree that these sentences are not well-written because we only focused on the long-time scale variations, so we used this comment to improve the following discussion of the comparison. Please see lines 397-403.

   Despite a lower resolution for MD77-191 *H. elegans* $Cd_w$ records, when compared to the $C_{org}$ and the *G. bulloides* percentage from core SK237 GC04, all of them seem to exhibit similar trends at the long-time scale from the last deglaciation to Holocene; however, some little discrepancies can be observed at millennial time scales, especially during the late Holocene (Fig. 5).

e) (minor point) Figure 6 needs a better embedding/explanation in the manuscript. Some of the records are neither explained in the main text nor in the figure caption.

Answer: Thanks for this reminding. We have detailed these records explanation in the revised version (lines 481-486) as the following paragraph, especially for figures 6 d-f records.

   These changes are consistent with the weakened the summer monsoon intensity, with less rainfall during the late Holocene, as observed in the BoB using core MD77-176 seawater $\delta^{18}O$ and core SO188-342KL $\delta D_{Alk-ic}$ records (Marzin et al., 2013; Contreras-Rosales et al., 2014; Figs. 6 e-f). In addition, this is also strongly supported by the $\delta^{13}C_{wax}$ records from the Lonar Lake over

the Indian continent (Sarkar et al., 2015; Fig. 6d) and a progressive increase in monsoon summer winds to the South of India (Bassinot et al., 2011).

Besides, in the figure and caption, we also made the brief description.

Overall, there are some useful data in this manuscript. The discussion of the data and subsequent interpretation lacks maturity at this stage and requires improvement. A moderate to major revision is required.

**Reply to reviewer #2 comments:**

We thank Reviewer Dr. André Bahr for providing helpful comments on our manuscript, they have been carefully considered. Please find below our answers to these comments.

Review of Changes in productivity and intermediate circulation in the northern Indian Ocean since the last deglaciation: new insights from benthic foraminiferal Cd/Ca records and benthic assemblage analyses by Ma et al.

The authors present benthic foraminiferal assemblage records and Cd/Ca data from the western Arabian Sea and Bay of Bengal (BoB) to investigate surface primary productivity (PP) and intermediate water mass variability in the context of the last deglaciation and Holocene climatic evolution. The authors find that Cd/Ca is primarily controlled by PP during the Holocene which mirrors monsoonal intensity. Notably, a strong monsoon is inferred to suppress PP in both areas due to enhanced run-off which increases upper ocean stratification in the BoB and reduced Ekman-upwelling off India as a result of decreased wind stress. During the deglaciation, the authors infer a dominance of water mass changes in driving the Cd/Ca signal, showing an enhanced advection of AAIW into the northern Indian Ocean during YD and HS1.

In general, the data and its interpretation appear mostly sound and in line with existing concepts about the paleoceanography of the Arabian Sea and the BoB as well as the influence of the monsoon on the PP in these areas. In this respect I find it noteworthy that the data (i) supports the presumed E-W dipole between strong upwelling off Oman and weak upwelling off India, and (ii) that maximum monsoon induced run-off in the BoB apparently suppresses PP due to strong stratification, despite riverine nutrient input should enhance plankton blooms on surface level. The authors mention stratification as an explanation rather briefly, however, I would encourage the authors to devote one or two more sentences on this issue (see also my detailed comment).

Answer: The reviewer suggests that the arguments for the monsoon intensity influence on the PP in the southeastern Arabian Sea and northeastern BoB during the Holocene should be stronger. In order to reinforce the discussion, we developed this part in section 5.3 (lines 447-451), improving the quality of the discussion about the contribution of nutrient inputs from rivers in these areas, even if this process seems not to be significant. We are now providing the following explanation in the manuscript:

However, the distribution of chlorophyll in surface water of the western BoB suggests a low annual productivity, indicating that the BoB is not significantly influenced by the riverine nutrient input (Zhou et al., 2020). Thus, it is likely that this increase in fresh water drove pronounced

ocean stratification in the northeast BoB, which could impede the nutrient transfer from deep layer to the euphotic upper seawater column, and then inducing low productivity.

I can also follow the arguments for the inferred intrusion of AAIW into the northern Indian Ocean during HS1 and YD, which agrees with the well-documented enhanced northward protrusion of this water mass in the Atlantic Ocean. However, I am not convinced by the way the authors come to this conclusion, which is based on the claimed mismatch between increasing *G. bulloides* abundances and decreasing Cdw estimates during YD and HS1. As depicted in the figure below, both records essentially follow the same trend, also bearing in mind that the resolution of Cdw is relatively low during YD and HS1. Hence, the proposed anti-correlation between *G. bulloides* abundances and Cdw seems to be an overstatement.

Irrespective of this problem, the good match of Cdw and $^{13}$C with AAIW reference records make it reasonable to assume that the Cdw values in deed capture water mass variability between HS1 and YD (Fig. 7). Hence, the interpretation at the end seems correct, but it is more likely that the relatively modest increase in PP during the YD/HS1 appear to have had a negligible influence of the Cdw. Only if PP is really high (such as in the mid-Holocene) Cdw is dominated by PP, as also indicated by the very high values > 1.0. The authors are somewhat over-confident regarding the use of Cd/Ca as a water mass tracer in the such potentially highly productive areas and might consider toning down their argumentation.

Answer: We fully agree with the reviewer that the conclusion - *H. elegans* Cd$_w$ displays an anti-correlated compared with the *G. bulloides* abundances during YD and HS1 - seems to be overstated because of the relatively low resolution during the last deglaciation. In order to take this comment into account, we slightly modified the discussion by separating the interpretation of the comparison between Cd$_w$ and *G. bulloides* abundances (revised section 5.4, lines 500-517). We are now providing the following corrections in the manuscript:

Significant decreases in *G. bulloides* relative abundance of cores SK237 GC04 (Naik et al., 2017) and MD77-191 records were observed from the HS1 to B-A (Bassinot et al., 2011), and thereafter slight increases occurred in the YD (Fig. 5). These high values at both core sites during the HS1 and YD may indicate an enhanced surface productivity during these intervals (Fig. 5). This should have led to increased intermediate Cd$_w$ and organic matter preservation under low oxygen concentration conditions during the HS1 and YD. However, despite a low resolution for the MD77-191 Cd$_w$ record during the last deglaciation, we do not observe high values of intermediate Cd$_w$ during the HS1 and YD (~0.6 nmol/kg) compared with the late Holocene (~1.59 nmol/kg), especially at 16.5-16 cal kyr BP. Although we cannot fully discard the influence of surface productivity on the intermediate Cd$_w$ in these time intervals, this apparent discrepancy seems to provide another evidence for the influence of changes in water masses and/or ventilation during the HS1 and YD, as already demonstrated by previous studies and proxies in the northern Indian Ocean (Bryan et al., 2010; Yu et al., 2018; Ma et al., 2019; 2020).

While the manuscript is well written, the Figures might benefit from rearrangement to make the discussion more easier to follow (cf. detailed comments below).

Given some moderate revisions I support publication of this study which represent an important contribution to our understanding of the deglacial evolution of the Indian Ocean.

[Figure]

Figure 5 with Cd/Ca from MD77-191 in blue, illustrating that Cd/Ca follows *G. bulloides*-abundances (even more in the more high-resolution data of adjacent core GC04).

Answer: Although the new figure suggested by the reviewer allows directly comparing the Cd/Ca with the % *G. bulloides* record, it may be difficult to decipher both records. Moreover, the reviewer superimposed records from two different cores, even if the global trends are very close to each other. Thus, we prefer to keep the Fig. 5 in its original version.

**Detailed comments**

Line 78: The motivation for the study is rather weak (essentially: we know little about the paleoproductivity of the BoB). It would be good to more explicitly state why we should care about this issue.

Answer: We agree with the reviewer and add some sentences (lines 78-87) to clarify the motivation as is shown in the following paragraph.

   By contrast, little is known about the paleoproductivity of the BoB, especially its links to changes in monsoon precipitation (Phillips et al., 2014; Zhou et al., 2020). Consequently, studying paleoproductivity and past nutrient concentration of intermediate water masses in the northeastern Indian Ocean will also allow us to completely understand the influence of monsoon climate changes in tropical ocean ecology at different timescales. Besides, as the benthic foraminiferal Cd/Ca is a promising proxy to reconstruct the intermediate-deep water nutrient content (e.g., Boyle and Keigwin, 1982; Tachikawa and Elderfield, 2002; Came et al., 2008; Poggemann et al., 2017; Valley et al., 2017), most of the studies referred to above have reconstructed deep-intermediate water masses in the past (e.g., Came et al., 2008; Bryan and Marchitto, 2010; Poggemann et al., 2017; Valley et al., 2017), and only few works indicate the relationship between the intermediate water masses nutrient and surface productivity (Bostock et al., 2010; Olsen et al., 2016).

L. 85: "estimate past changes in the nutrient content, since the last deglaciation, over the last 17 kyr BP." The last part is redundant.

Answer: We have corrected this sentence in the revised manuscript (line 93) to be:

"estimate past changes in the nutrient content since the last deglaciation."

l. 100: "of the planktonic…"

Answer: It has been done. Please see line 108.

l. 101 (and elsewhere): avoid using "." as multiplicator

Answer: It has been done. Please see lines 109-110.

L. 109: "Arabian High Salinity Waters" (all capitals)

Answer: Corrected. Please see line 117.

L. 118: Neither Fig. 1 nor S1 show salinity.

Answer: We fully agree this comment and removed it in the revised manuscript. Please see line 127.

L. 131: "northern intermediate …" (no capitals)

Answer: Corrected. Please see line 140.

Section 3: Please also include the statistical methods used in the study in the Methods chapter. Which program did you use to perform the PCA? Did you use a correlation or variance/covariance matrix?

Answer: We have corrected this sentence in the revised manuscript (lines 178-179) to be:

In order to describe major faunal variations, we performed principal component analysis (PCA) on the variance-covariance matrix using the PAST software (Version 3.0, Hammer et al., 2001).

Section 3.1.: Regarding the design of the study, I wonder why the authors decide to use four different species, when *H. elegans* is available as a well-documented, faithful recorder of bottom water Cd/Ca. What was the rationale to use the three calcitic species, especially as they include infaunal dwellers which are naturally not the best suited for detecting bottom water fluctuations?

Answer: We fully agree with the reviewer that *H. elegans* is a well-documented and faithful recorder of bottom water Cd/Ca. Indeed, Tachikawa and Elderfield (2002) indicated that due to the lower partition coefficients, the infaunal benthic foraminifera could record Cd/Ca values similar to *Cibicidoides*, despite elevated pore water Cd. Thus, many previous studies used both *Cibicidoides* and *Uvigerina* in paleoceanographic reconstructions (e.g., Marchitto and Broecker, 2006; Makou et al., 2010; Umling et al., 2018; 2019). At core MD77-191 site, we could provide continually calcite benthic species samples with different microhabitat (*Cibicidoides pachyderma*, *Uvigerina peregrina*, and *Globobulimina* spp.). Therefore, we performed Cd/Ca analyses on these benthic species to improve understanding of possible species level differences and microhabitat effects on the benthic Cd/Ca records. We clarified this point in the revised manuscript (lines 149-152) as:

In order to improve understanding of possible species level differences and microhabitat effects

on the benthic Cd/Ca records, we analyzed Cd/Ca in three calcite (*Cibicidoides pachyderma*, *Uvigerina peregrina*, and *Globobulimina* spp.) and one aragonite (*Hoeglundina elegans*) benthic foraminiferal species from core MD77-191.

L. 204-205: you might omit "over the last deglaciation"; add an "a" before "significant decrease"
Answer: We have corrected this sentence in the revised manuscript. Please see lines 214-215.

L. 228 etc., regarding the PCA results: You show 2 PCs which explain 61% of the total variance. What's about the other PCs, how much variance to they explain and what was the rationale to limit the investigations to those two PCs?
Answer: The total variance of other PCs is 49% which is shown in the following figure. Compared with the total variance of PC1 (42%) and PC2 (19%), PC3 is the largest one and only explains 8% of the total variance for the rest PCs. The species composition consists of *Hoeglundina elegans* (0.66), *Globobulimina* spp. (0.22) (Positive loadings), *Uvigerina peregrina* (-0.59), *Cibicidoides pachyderma* (-0.21) (Negative loadings). It seems that the main composition of assemblages (PC3) is quite similar to PC1 and does not show more information about the bottom conditions. Thus, we only use PC1 and PC2 in the manuscript to recognize the three assemblages.

[Figure]

Figure: the variance of total PCs for core MD77-191

When the authors discuss the loadings of the individual PCs, they assign particular species with a very low loading to individual assemblages. PC1 for example is very much dominated by *B. aculeata* (+0.84 loading); the denoted loadings of -0.07 and less for *B. manginata*, *C. wuellerstorfi*, *G. subglobosa* (Table 1) appear to be rather insignificant. The same applies for PC2 which has high loadings of +0.42 and -0.62 for *S. bulloides* and *H. elegans*, respectively; I doubt that e.g. *G. soldanii* with a loading of 0.07 has a significant relevance to PC 2. Please reconsider the discussion of the PC 1 and PC 2 accordingly. You might also consider providing a bi-plot for PC 1 and PC 2 as an extra figure.
Answer: We agree that the dominant species (*B. aculeata*, *S. bulloides* and *H. elegans*) make a significant contribution for these three assemblages, respectively. We recognized three benthic assemblages based on the positive and negative loadings of different PCs. Despite the loading values of *B. manginata*, *C. wuellerstorfi* and *G. subglobosa* are much less compared with the dominant species (*B. aculeata, S. bulloides and H. elegans*), both these lower loadings species are environmental sensitive species, associated with different bottom water conditions (e.g., Corliss et al., 1986; Schmiedl et al., 1998; Almogi-Labin et al., 2000). Thus, it seems reliable to use these benthic species for the interpretation of assemblages. In addition, we have plotted the PC1 and

PC2 records together in Figure 3, so we prefer not to add an extra figure about the bi-plot for PC 1 and PC 2 in the manuscript.

L. 249: "aragonite": change into "argonitic"
Answer: It has been done. Please see line 259.

L. 262: The paragraphs discussing Cdw repeat in large parts what have been written about the Cd/Ca ratio in the Results chapter. Please avoid such duplication. The same also applies for Figures 2 and 4. Figure 2 might be moved into the supplement.
Answer: We agree with the reviewer that the description of $Cd_w$ records is similar with the Cd/Ca ratios, so we modified and removed some sentences in the section 5.1 about the discussion of intermediate water $Cd_w$ results from the Northern Indian Ocean to avoid the duplication. We are now providing the following descriptions in the manuscript (lines 274-322):

The intermediate $Cd_w$ results based on the *H. elegans* Cd/Ca values of core MD77-191, range from 0.5 to 3.1 nmol/kg since 17 cal kyr BP (Fig. 4a), with a core top value of 0.80 nmol/kg in agreement with the estimated intermediate water depth modern $Cd_w$ (~0.83 nmol/kg) in the northern Indian Ocean (Boyle et al., 1995). The intermediate $Cd_w$ was also calculated from calcite benthic species *C. pachyderma*, *U. peregrina* and *Globobulimina* spp. from core MD77-191, with values ranging between 0.53-1.48 μmol/mol, 0.52-1.04 μmol/mol and 0.26-0.65 μmol/mol, respectively (Fig. 4a). The $Cd_w$ values of *C. pachyderma* and *U. peregrina* are within the same range. However, the *H. elegans* $Cd_w$ values are higher than those from the two calcite species, especially during the Late Holocene. Moreover, the core top data of *C. pachyderma* and *U. peregrina* are also lower (~ 0.7 and 0.69 nmol/kg, respectively) than the modern estimated $Cd_w$ data (~ 0.83 nmol/kg) in the northern Indian Ocean (Boyle et al., 1995) (Fig. 4a). These depleted $Cd_w$ values may be related to the benthic foraminiferal microhabitat effect; indeed, *U. peregrina* is known to be strictly a shallow infaunal species, as well as *C. pachyderma* (Fontanier et al., 2002), differing from strictly epifaunal taxa, such as *Cibicidoides wuellerstorfi* (Mackensen et al., 1993).

Besides, the deep infaunal *Globobulimina* spp. $Cd_w$ displays relatively much lower values and does not exhibit strong variations compared to the other species investigated in this study, displaying a general increasing trend from the last deglaciation to the Holocene. As *Globobulimina* spp. correspond to deep benthic infaunal species, this result may indicate a stable nutrient content of pore water, as compared to other benthic taxa associated with bottom water (Fig. 4a). Thus, when tracking past changes in the bottom water $Cd_w$ concentrations, the use of a strictly epifaunal species living at the water-sediment interface such as *H. elegans* appears to be more robust than using endofaunal species that live in contact with pore water.

Relative variations in the $Cd_w$ obtained from *C. pachyderma* and *U. peregrina* are in good agreement with the records obtained on *H. elegans*. Variations of *H. elegans* $Cd_w$ during the last deglaciation indicate a decrease of about ~0.6 nmol/kg in the HS1 and YD periods, with a slight increase (0.9 nmol/kg) during the warm B-A. $Cd_w$ results from core MD77-191 indicate a shift from the last deglaciation (~0.7 nmol/kg) to the late Holocene (~1.59 nmol/kg). During the Holocene, the $Cd_w$ records display relatively low values of around 0.9 nmol/kg in the 10-6 cal kyr BP time interval, and show a major shift at around 6.4 cal kyr BP with values rising up to 3.1 nmol/kg.

The reviewer also suggests that Figure 2 might be moved to the supplementary information, but

we do not fully agree with this comment. Indeed, comparing the *Globigerinoides ruber* $\delta^{18}O$ records of cores MD77-191and MD77-176 with GISP2 Greenland ice core $\delta^{18}O$ signal (Stuiver and Grootes, 2000) in Figure 2 allows defining the key time intervals we focus, i.e. the Heinrich Stadial 1, the Younger Dryas, the Bølling-Allerød events and the EHCO. We think that it helps to clearly describe the results (Cd/Ca and benthic assemblages), as well as the following discussion. There, we prefer to use Figure 2 here rather than move it to the supplement.

L. 264: please add the modern Cdw to figure 4 for reference.
Answer: We improved figure 4 with this common. Please see new Figure 4.

[Figure]

**Fig. 4.** (a) $Cd_w$ records calculated based on the Cd/Ca of benthic foraminifera *Hoeglundina elegans* (black), *Cibicidoides pachyderma* (green), *Uvigerina peregrina* (blue), and *Globobulimina* spp. (orange) obtained from core MD77-191, (b) $Cd_w$ record from core MD77-176 reconstructed using *H. elegans* Cd/Ca, the red line is the smoothed curves using a five-point average. The red stars represent the modern $Cd_w$ (~0.83 nmol/kg) in the northern Indian Ocean (Boyle et al., 1995). The color shaded intervals and abbreviations are the same as in Figure 2.

Figures 2 and 4: The y-axes should have a common scale to enable direct comparison of the individual records. As shown, the Cd/Ca and Cdw records of *Globobulimina* spp. appear to show large fluctuations, however, compared to the other species these fluctuations are of rather minor importance (as stated in the text).
Answer: We provide new figures with a common scale for y-axes. Please see new Figure 2. The new Figure 4 is already shown in the reply for the above question "L. 264".

[Figure]

**Fig. 2.** (a) GISP2 Greenland ice core $\delta^{18}O$ signal (Stuiver and Grootes, 2000). (b)-(c) *Globigerinoides ruber* $\delta^{18}O$ records of cores MD77-191and MD77-176, respectively (Marzin et al., 2013; Ma et al., 2020). (d) Cd/Ca records of the benthic foraminifera *Hoeglundina elegans* (black), *Cibicidoides pachyderma* (green), *Uvigerina peregrina* (blue), and *Globobulimina* spp. (orange) obtained from core MD77-191; (e) Cd/Ca records of the benthic foraminifera *H. elegans* from core MD77-176. EHCO for Early Holocene Climate Optimum, YD for Younger Dryas, B-A for Bølling-Allerød and HS1 for Heinrich stadial 1.

L. 286: the core tops values mentioned here should be shown in the respective Figure for reference.
Answer: Corrected. Please see new Figure 4 in the reply for question "L. 264: please add the modern Cdw to figure 4 for reference".

L. 300-303: The final statement that *H. elegans* provides the most reliable Cd/Ca (or Cdw) data is not really surprising. As stated in the earlier comment I would appreciate if the authors could provide more arguments what they wanted to test/proof with including the other three species.
Answer: Please refer to the reply for question "Section 3.1.: Regarding the design of the study, I wonder why the authors decide to use four different species, when *H. elegans* is available as a well-documented, faithful recorder of bottom water Cd/Ca. What was the rationale to use the three calcitic species, especially as they include infaunal dwellers which are naturally not the best suited for detecting bottom water fluctuations?"

Indeed, we indicated in the original manuscript in lines 270-277 (revised version, lines

285-289), which discussed deep infaunal *Globobulimina* spp. $Cd_w$ results. Lines 286-293 (revised version, lines 274-284), we compared $Cd_w$ obtained from *C. pachyderma* and *U. peregrina* with records obtained on *H. elegans*, the depleted $Cd_w$ values of calcite infaunal species may be related to the microhabitat effect, thus *H. elegans* appears to be more robust than using endofaunal species that live in contact with pore water. Although we modified the sentences and structure of the discussion about intermediate water $Cd_w$ results in the section 5.1 to avoid the duplication, we still keep the comparison of these four benthic species $Cd_w$. Please refer to reply for question "L. 262: The paragraphs discussing $Cd_w$ repeat in large parts what have been written about the Cd/Ca ratio in the Results chapter. Please avoid such duplication. The same also applies for Figures 2 and 4. Figure 2 might be moved into the supplement".

L. 307 etc.: I think the presentation of the Assemblage data could be improved: 1) as stated earlier the presentation of the PCA results is not totally convincing;
Answer: As mentioned before (please refer to question Line 228), the total variance of PC1 (42%) and PC2 (19%) could explain 61% of the total variance, and for other PCs, the total variations are much less. Besides, the main composition of the rest PCs negative or positive loadings is dominated by the same benthic species as these recognized three assemblages (such as based on PC3, 8% total variation, the largest one among the rest PCs), it is difficult to glean more additional information from this regarding bottom conditions. Thus, it seems reliable that we recognize three assemblages in this paper.

2) Why do the authors start with Assemblage 3 not in the numerical order?
Answer: We interpreted our records in time order from the last deglaciation to Holocene.

3) If they use the Assemblages as environmental indicator they could plot the abundance of the respective Assemblages instead of individual foraminifera species in the results figure; they could also assign specific environmental parameters to each Assemblage in the Figures (e.g. Assemblage 1 = more/less productivity or oxygenation), which would help to more concisely convey the message of the study.
Answer: The two first-ranked principal components are selected to obtain species associations with demands to specific environmental conditions. Based on the positive and negative loadings of PC1, we recognized benthic assemblages 1 and 2, respectively. In addition, assemblage 3 was identified by the positive loadings of PC2. And then we explored to discuss the bottom water environmental condition changes, combining different major individual benthic taxa of the assemblages, thus, the abundance of respective assemblages may be not suitable as the environmental indicator. Therefore, we prefer to keep the figures in their original version.

L. 351: better write "Assemblage 1" instead of "fauna 1"
Answer: Corrected. Please see line 384.

L. 353. "depleted *Globigerina bulloides* abundances" – replace by "low *G. bulloides* abundances"
Answer: It has been done. Please see line 386.

L. 370: please refer more often to the respective Figures.

Answer: We have added the reference figures 3 and S2 in the revised version. Please see line 412.

L. 398: a decreasing Cdw trend between 5.2 and 2.4 cal kyr BP is not evident for me, the values are constantly very high during this time period.

Answer: We agree with the reviewer that the $Cd_w$ values are constantly high during 5.2-2.4 cal kyr BP, especially compared with the early Holocene. Thus, we have corrected this description in the revised version (line 440) as:

"reach a maximum during the late Holocene".

L. 405-406: the relation of stratification and PP should be discussed in more detail (see my general remark).

Answer: We used this comment to improve the discussion of the manuscript. We are now providing the following explanation in the manuscript (lines 447-451):

However, the distribution of chlorophyll in surface water of the western BoB suggests a low annual productivity, indicating that the BoB is not significantly influenced by the riverine nutrient input (Zhou et al., 2020). Thus, it is likely that this increase in fresh water drove pronounced ocean stratification in the northeast BoB, which could impede the nutrient transfer from deep layer to the euphotic upper seawater column, and then inducing low productivity.

L. 431-432: The authors refer to summer insolation – pleas show it in an appropriate figure (e.g. Fig. 5)

Answer: Bassinot et al. (2011) indicated that the ITCZ location shifted northward when boreal summer insolation reached a maximum in the early Holocene, associated with the enhanced summer monsoon wind intensity and a decrease in the Ekman pumping in the southern tip of India. This configuration then induced a decrease in surface productivity in the southeastern Arabian Sea. We added the summer insolation variations in the modified Figure 6, please see the new Figure 6.

[Figure]

**Fig. 6.** (a) The solar insolation at 10°N in summer (Laskar et al., 2004). (b) and (c) intermediate $Cd_w$ calculated from *H. elegans* obtained from MD77-176 and MD77-191, respectively. (d) Lonar Lake $\delta^{13}C_{wax}$ record (Sarkar et al., 2015). (e) $\delta D_{Alk-ic}$ record from core SO188-342KL (Contreras-Rosales et al., 2014). (f) Seawater $\delta^{18}O$ anomaly obtained from MD77-176 (Marzin et al., 2013). The color shaded intervals and abbreviations are the same as in Figure 2.

L. 453: The reference to Figure 7: should't it be rather Fig. 6?
Answer: Corrected. Please see line 500.

L. 457-458: "Thus, we do not expect that surface productivity played an important role during the last deglaciation." This statement is odd, as it has been discussed at great length that PP is influencing Cd/Ca. The following "In addition,…" does also not fit as the following sentence does not support the above notion of PP playing an unimportant role.
Answer: We agree with the reviewer that these sentences seem to be odd here and removed them in the revised version. Please see lines 505-506.

L. 462 etc: I am not convinced by the statement that increasing *G. bulloides* abundances during HS1 and YD are in conflict with both, Corg and Cdw records. With regards to Corg I agree that it declines opposite to the trend in *G. bulloides*, however, Corg does not only depend on PP but also on preservation, and potentially sedimentation rate (one way to check the influence of sediment accumulation would be to compute Corg accumulation rates). However, Cdw rather follows *G. bulloides* abundances, at least it is not anti-correlated, as one might infer from the text.
Answer: We agree with the reviewer and correct these sentences in the revised version. We also fully agree with the reviewer that the $C_{org}$ could depend on PP and/or preservation, this has been also indicated in the revised manuscript (lines 391-393). In this study, we mainly focus on the *G. bulloides* abundance to reflect the paleoproductivity in this region. The modifications of revised lines 462 etc are explained in Reply the second general comment). Please see lines 500-517.

L. 482-487: the sentence is too long and complicated, please rephrase.
Answer: We have corrected this sentence in the revised manuscript (line 535-540) to be:
   Thus, as the benthic $\delta^{13}C$ values collected from the north Indian Ocean could better constrain the influence of AAIW in the two studied cores (Naqvi et al., 1994; Jung et al., 2009; Ma et al, 2019; 2020), we can also compare the range values of AAIW $Cd_w$ from both studied cores with data from Atlantic and Pacific Oceans at intermediate water depth during the HS1 and YD ($Cd_w$, 0.3-0.9 nmol/kg; Umling et al., 2018; Valley et al., 2017).

L. 496: "increase" instead of "icrease"
Answer: Corrected. Please see line 549.

L. 504: "the entire biological factory was related to reduced monsoon intensity" – see my general comment: this statement need more justification. The presence/absence of *G. bulloides* might well be influenced by stratification and surface water freshening, but does this apply to other primary producers as well?
Answer: Unfortunately, we have no other productivity indicators records available from core

MD77-191 (Arabian Sea).

**General remarks to the figures:**

It helps the reader if the authors state next to the core name where the core is located (e.g. within the BoB)

Answer: For figure 1, we have put the "Arabian Sea" and "BoB" names in the right place. And for these two figures, there are more information should be shown clearly, such as the general surface circulation and primary productivity distribution of the northern Indian Ocean. If we move the "Arabian Sea" and "BoB" names next to the core name, these two figures seems to be "confused", so we prefer to keep the core name in its original version.

Fig. 5: you might add here the benthic $^{13}$C records of MD77-191 and -176 used to discussed water mass variability (cf. Fig. S3)

Answer: We do not fully agree with this comment since for Fig. 5, in order to examine the relationships between primary productivity and intermediate $Cd_w$ records of the studied cores, we compare MD77-191 $Cd_w$ results with different records associated to surface productivity from the same area (southeastern Arabian Sea). The similarity of the benthic $\delta^{13}$C-increases reflects the northward expansion of AAIW during the last deglaciation in the north Indian Ocean and Pacific Ocean (Pahnke and Zahn, 2005; Jung et al., 2009; Ma et al., 2019, 2020), the detailed interpretation based on the benthic $\delta^{13}$C records (including cores MD77-191 and MD77-176) have been discussed in these previous papers, thus we prefer to use Fig. S3 as the supplementary information to interpret the changes of intermediate water masses.

S1: please add the variable + unit to the color shading on the right side of the maps.

Answer: Corrected. Please see new Figure S1.

[Figure]

**Fig. S1.** a) – b) The Net primary productivity distribution in the Northern Indian Ocean during January and July, respectively. Maps based on MODIS chlorophyll-a, SST, PAR satellite data,

using the standard vertically Generalized Production Model (VGPM) (Behrenfeld and Falkowski, 1997) as the standard algorithm.

**Author response: version 2**

Dear editor,

Thank you for your consideration and all the helpful comments. We have carefully revised the manuscript according to the comments from you and all reviewers. The answers and explanations are in blue color, the corresponding change in the revision-marked version was tracked in red color. The point-to-point responses to the editor and reviewers are listed as following:

**Response to the Editor:**

1. There need to be more a consistent interpretation of changes in the $Cd_w$ values (as the reviewer already mention in the first review).

Answer: We fully agree with editor and reviewer #1 that the interpretation of $Cd_w$ records should be more consistent. We have enhanced the discussion by the evaluation of all possible factors on intermediate $Cd_w$ records at different time scales. Please see the revised lines 421-444 (section 5.2):

From the last deglaciation to the late Holocene, the $Cd_w$ record displays a significant shift from ~0.7 nmol/kg to about twice values of ~1.59 nmol/kg. The intermediate $Cd_w$ values are thus extremely high during the late Holocene and synchronous with the higher values of $C_{org}$ and *G. bulloides* percentage records. These observed similar trends suggest that the increased surface productivity at the core site during the late Holocene is associated to higher intermediate $Cd_w$ values. Besides, previous studies have suggested that increased $Cd_w$ values (>1 nmol/kg) could correspond to elevated surface productivity (Bostock et al., 2010; Olsen et al., 2016). However, at millennial time scale, we also observed several decreases in intermediate $Cd_w$ values (~0.81 nmol/kg) during the late Holocene, reaching nearly similar values during the last deglaciation (Fig. 5). Thus, the variations in the $Cd_w$ values cannot be fully associated to variations in the surface productivity.

As mentioned before, during the Holocene, an increased influence of NADW in IDW was observed in the northern Indian Ocean (Yu et al., 2018; Ma et al., 2019; 2020). NADW is characterized by a depleted nutrient content (modern $Cd_w$, ~0.2 nmol/kg; Poggemann et al., 2017), and its contribution to IDW may affect the intermediate $Cd_w$ by deep-water masses upwelling when flowing northward. However, during the late Holocene, benthic foraminiferal assemblage 1 is associated to lower oxygen concentrations, which seem to be inconsistent with an enhanced influence of better ventilated NADW in IDW in the northern Indian Ocean. Therefore, this appearing discrepancy seems to indicate that deep-intermediate water masses variations is not an important control during the Holocene in this area, although we could not fully exclude the influence of NADW in IDW at millennial time scale. Moreover, there is no clear evidence for such a millennial-scale variability in the IDW and/or NADW circulation in the studied area. Thus, we suggest the intermediate $Cd_w$ at core MD77-191 site may be mainly influenced by surface productivity, especially during the Holocene.

2. I consent with reviewer that the depth of NADW and IDW in the Indian Ocean needs to be clarified be clarified according to the reviewers comment and the literature. In order to verify the water mass interpretation, a) the hydrography section still needs further improvement and b) there needs to be a more in-depth explanation how of a deep-water mass, currently occurring below ~2km, affect sediment cores at true intermediate water depth.

Answer: We fully agree with the reviewer that the discussion about the NADW has to be improved. We have added a short paragraph to introduce the detailed information on modern hydrological setting at the studied site. Please see the revised lines 130-137 (section 2):

Due to the land-sea configuration in the north by Asia, the deep waters of the northern Indian Ocean originate from the south, including the Circumpolar Deep Water (CDW) and North Atlantic Deep Water (NADW) (You, 2000; Tomczak and Godfrey, 2003; Talley et al., 2011). Thus, between 1500 and 3800m, the dominant deep water in the North Indian Ocean is Indian Deep Water (IDW), originating from the CDW admixed with NADW (You, 2000; Tomczak and Godfrey, 2003; Talley et al., 2011). Then, during their pathway, the bottom water upwells when it expands northward in northern Indian Ocean, returning to shallower depths (You, 2000, Figure 1c). Therefore, variations of deep water masses can also influence the intermediate-depth waters in the northern Indian Ocean.

Besides, to be more precise, we also revised the use of "NADW" as "enhanced contribution of NADW in IDW" in the discussion. Please see lines 361, 363, 380, 391, 432, 437 and 440.

**Reply to reviewer #1 comments:**

Review of manuscript "Changes in productivity and intermediate circulation in the northern Indian Ocean since the last deglaciation: new insights from benthic foraminiferal Cd/Ca records and benthic assemblage analyses" by Ma et al.. This is a revised version of a manuscript that presents data from a sediment cores from the northern Indian Ocean (Arabian Sea and Bay of Bengal), comprising geochemical time series generated on benthic foraminifera as well as census data for planktic and benthic foraminifera. Based on these data, the authors deduce monsoon driven changes in productivity mainly dominating the various records during the Holocene and changes in intermediate water chemistry during the deglaciation. Whilst the revised version of the manuscript has improved by addressing some of the issues raised, there are still a number of problems hampering wholehearted support at this stage.

These are some of the issues (line references are based on the "...ACT1" version of the manuscript that highlights changes in the document):

a) One issue still relates to the consistency of interpreting the Cdw records (a part of this section is unchanged text from the previous review). In chapter 5.4 the authors claim that the Cdw values during the deglaciation are lower than during the Holocene. First, this statement is only correct if longer term averages are considered. On short time scales (which need to be considered, given that this is a chapter on millennial scale change), the youngest Cdw data in core MD77-191 (2-1.5 KaBP) are comparable to the low YD and HS1 values. Up to this point a big effort has gone into establishing Cdw as reflecting productivity variations at the sea surface and the related flux of organic carbon. Now the focus shifts to bottom water ventilation changes being recorded. If general water ventilation would play are role in setting the recorded Cdw values, this has to apply to the Holocene too and would therefore need to be considered there. The revised version of the

text does contain some extra lines, but the arguments made are rather unconvincing. First, the discrepancies between the Cdw and the *G. bulloides*/Corg data in figure 5 during the Holocene are not "little" as stated in line 400. There seems to be some covariation at times, but there are also significant offsets during other periods, in particular in the youngest part of the record. Both, *G. bulloides* records in figure 5 as well as the Corg record suggest that the there is high productivity in the latest part of the Holocene. Yet, the Cdw values in core MD77-191 drop back to very low values. Taking this observation as face value, the long-terms trends between the records do show differences too. The authors then argue that the Cdw values in the late Holocene are too high (quoting a value of 1.59 nmol/kg which is difficult to reconcile with figures 4 and 5) to be explained by water mass related changes (here discussed as NADW influence). This refers back to the Cdw record of core MD77-191, however, which does show very low values in the top section, which would be in line with a low Cdw water mass incursion and are not consistent with the high productivity values indicated in the other data in figure 5 for the latest Holocene. This part of the discussion should contain a more in-depth evaluation of all possible factors explaining the Cdw record. In addition, ignoring (too a large extend) the millennial scale change in the Holocene in Cdw records may turn out to be a missed chance.

Answer: To take into account the comments from reviewer #1, first, we rephrase each time necessary in the manuscript the "little millennial timescale variations" (See, for instance, Lines 420, 428, 440 and 475). Then, the reviewer indicates that the interpretation of $Cd_w$ records should be more consistent, and need to contain the evaluation of all possible factors, especially on a millennial time-scale during the late Holocene (2-1.5 cal kyr BP). In order to clarify this point, we developed the discussion in section 5.2 (lines 421-444) about the detailed discussion of all the possible factors on the intermediate $Cd_w$ at different time scales. Briefly, at a long-time scale, a significant increased trend was observed for MD77-191 $Cd_w$ records from the last deglaciation to the late Holocene. These extremely high values during the late Holocene are consistent with the increased $C_{org}$ and *G. bulloides* percentage records, as well as changes in the benthic assemblages, associated with the increased surface productivity at the core site during the late Holocene. Besides, previous studies have suggested that increased $Cd_w$ values (>1 nmol/kg) could correspond to elevated surface productivity (Bostock et al., 2010; Olsen et al., 2016). All these records seem to indicate the important influence of surface productivity on intermediate $Cd_w$ during the late Holocene.

However, at millennial time scale during the late Holocene, we could also observe quite low intermediate $Cd_w$ values (~0.81 nmol/kg) in 2-1.4 cal kyr BP, which are similar with the values during the last deglaciation. Based on previous studies, increased influence of NADW in IDW was observed during the Holocene in the northern Indian Ocean (Yu et al., 2018; Ma et al., 2019; 2020). NADW is characterized by the depleted nutrient content (modern Cdw, ~0.2 nmol/kg; Poggemann et al., 2017), may affect the intermediate $Cd_w$ by deep-water masses upwelling when flowing northward. However, during the late Holocene, benthic foraminiferal assemblage 1 indicates the lower oxygen concentrations, that seem to be inconsistent with previous studies suggesting an enhanced influence of better ventilated NADW in IDW in the northern Indian Ocean. Therefore, although we could not exclude the influence of NADW in IDW at millennial time scale during the late Holocene, this appearing discrepancy cannot insure that deep-intermediate water masses variations played an important role during the Holocene in this area. Thus, we suggest that our initial interpretation could be also maintained for the Holocene,

even if we developed the discussion as suggested by Reviewer #1.

b) Usage of term NADW: In response to my previous comment on the referral to NADW, the authors did change the manuscript. It has improved to some degree but contains a misleading use of NADW. Initially NADW is introduced as a contributing water mass to IDW. The first problem is that usage thereafter only mentions NADW, whilst, at most, I guess, it would reflect an added contribution to IDW (?). Also, it is claimed that IDW would occur in the northern Indian Ocean between 1500 and 3800m (according to Talley 2011). Based on the usage of NADW in latter parts of the manuscript the reader is led to believe that NADW would also be occurring at water depths as shallow as 1500m. Yet, using the salinity maximum, referred to in Talley (2011, chapter 11, figure 11.16), as an indicator, NADW occurs below roughly 2000m, in line with the text in that chapter. In addition, the only region original (largely unmixed) NADW occurs is off the southeast coast of Africa, with the northward propagation blocked by the Davie Ridge (although there is some discussion in relation to a potential northward spillover occurring). In order to substantiate their argument, a) the hydrography section still needs further improvement and b) there needs to be a more in-depth explanation how (even contributions) of a deepwater mass, currently occurring below ~2km, affect sediment cores at true intermediate water depth. The changes in the revised version are not entirely convincing. This affects large parts of the discussion.

Answer: The reviewer suggests that the hydrography section should be improved, especially the explanation about the influence of deep waters at intermediate water depth in the northern Indian Ocean. In order to reinforce the interpretation, we have added a paragraph in the revised manuscript (section 2, lines 130-137), greatly improving the quality of the description of modern hydrological setting at the studied site. To do that, we especially based our description on a new reference: You, Y.: Implications of the deep circulation and ventilation of the Indian Ocean on the renewal mechanism of North Atlantic Deep Water, Journal of Geophysical Research: Oceans, 105, 23895–23926, 2000. Briefly, in the northern Indian Ocean, IDW lies between 1500 and 3800 m, originated from Circumpolar Deep Water admixed with NADW (You, 2000; Tomczak and Godfrey, 2003; Talley et al., 2011). As the deep water upwells when it moves northward, the deep waters can eventually return to shallower depths (You, 2000). Thus, changes in the deep waters can also affect shallower-depth water masses in the northern Indian Ocean.

Besides, we would also like to thank the reviewer for the comment, pointing out that the use of NADW in the discussion is not precise. We have replaced that with "enhanced contribution of NADW in IDW". Please see lines 361, 363, 380, 391, 432, 437 and 440.

c) The usage of PCA's, assemblages and/or individual species concentrations is rather confusing, and pieces of information are missing. In lines 238-239, e.g. the authors state that the PCA analysis yielded 3 factors, referring to table 1. Yet table 1 only shows 2. In lines 246-248, referral is made to assemblage 3 (PC3) without figures 3 and S2 actually showing PC3 scores. This should be clarified. In addition, the usage of assemblage and individual species concentrations is rather confusing. Given that the factor analysis did produce groupings of species, it would help to add plots showing change in the grouped assemblages. It would help the flow of the text and may add robustness to the results. This requires changes across the manuscript.

Answer: The confusion seems to come from the number of assemblages (3) compared to the use of 2 PCs. Thus, as indicated in section 4.2 (lines 250-279), we clearly clarified that we only use PC1 (positive and negative loadings) and PC2 (positive loadings) in the manuscript to recognize three assemblages. These two PCs could represent about 61% of the total variance, and these three

assemblages are dominated during the last deglaciation, early and late Holocene, respectively.

For PC3, as is shown in the following figure, compared with the total variance of PC1 (42%) and PC2 (19%), PC3 is the largest one and only explains 8% of the total variance for the rest PCs. The species composition consists of *Hoeglundina elegans* (0.66), *Globobulimina* spp. (0.22) (Positive loadings), *Uvigerina peregrina* (-0.59), *Cibicidoides pachyderma* (-0.21) (Negative loadings). It seems that the main composition of assemblages (PC3) is quite similar to PC1 and does not show more information about the bottom conditions. Therefore, we only focus on the PC1 and PC2 in the manuscript for the interpretation and do not present other PCs in the discussion. We added these explanations in section 4.2 (lines 255 to 260) and also put PC3 loadings in Table 1.

[Figure]

Figure: the variance of total PCs for core MD77-191

d) (somewhat minor point) Lines 543/4 Figure 7 needs a much better embedding in manuscript. This is first and only time it is mentioned. There is no real stipulation of the main findings. The reader to left to deduce this alone.

Answer: We used this comment to improve the discussion of Figure 7 by adding one sentence to detailed describe this figure in the revised version (Please see lines 571-574).

Detailed comments:

Lines 35-36 the ages in Monnin are not in line with this statement

Answer: We have corrected this mistake in the revised manuscript. Please see lines 42-43.

line 93 this statement is not true. The figures reach back 17-18 ka BP in most cases, which includes the deglaciation

Answer: We understand that the use of "since" appears confusing of Reviewer #1 to indicate the time period covered by our study. However, as the time interval of core MD77-191 (Arabian Sea) is from 17 to about 1 cal kyr BP, and the data obtained from MD77-176 (northeast BoB) range between 18 and 1 cal kyr BP, all of these data cover the interval of the last deglaciation to the Holocene. Thus, we prefer to keep the expression "since the last deglaciation" in the manuscript, which means these continuous records include both the last deglaciation and Holocene periods.

chapter 4.1: This chapter is not placed correctly. It would be better placed in the methods section.

Answer: In the literature about elemental ratios in foraminifera, it is classical to begin the description of the results by discarding the influence of contaminants thanks to Al/Ca, Mn/Ca and Fe/Ca ratios, so we just apply this common use from previous studies. Moreover, as these ratios are measured at the same time than the other elemental ratios, they can be described in the results

part. However, in order to take into account this comment, we added some sentences to interpret the reason for performing Mn/Ca, Fe/Ca and Al/Ca analyses in the method (Section 3.1, see lines 166-169).

Line 248-250 This statement is inconsistent with figure 3. The figure clearly shows that the statement only applies to parts of the Holocene and is not valid for the entire period. Also, the description of the counts is focused on the Holocene whilst earlier description cover the entire 18ka. This leaves the reader very confused as to the main thrust of the manuscript.

Answer: As we have detailed interpreted the benthic foraminiferal assemblage analyses in section 4.2, three assemblages were recognized based on the positive loadings (assemblage 1, during the late Holocene), negative loadings (assemblage 2, during the early Holocene) of PC1, as well as positive loadings of PC2 (assemblage 3, during the last deglaciation). Therefore, these three assemblages have covered the entire record from the last deglaciation to Holocene. In order to easy compare these three assemblages, we prefer to show the percentages of major species for three assemblages in figure 3, and put the rest associated species in the supplementary figure S2.

Besides, *Bulimina aculeate* and *C. pachyderma* dominate assemblage 1, which corresponds to the benthic foraminiferal fauna during the late Holocene. By contrast, *H. elegans* dominate assemblage 2, which is more important during the early Holocene. Besides, in figure 3, we can also observed high percentages of these species in late and early Holocene, respectively. Thus, we described in the manuscript that "However, the main species from negative loadings consist of *Bulimina aculeata*, *H. elegans* and *C. pachyderma*, which dominated the Holocene." It seems that the main composition of PC2 negative loadings is quite similar to assemblages 1 and 2, and then does not show more information. Therefore, we do not use the negative loadings of PC2 in the manuscript to recognize more assemblages.

Lines 337-338 this does not make sense. There is confusion in the use of "since". This includes the title.

Answer: Please refer to the reply for question "line 93 this statement is not true. The figures reach back 17-18 ka BP in most cases, which includes the deglaciation". As already detailed answer to the upper comment, the meaning of phrase "since the last deglaciation" is the interval of last deglaciation and Holocene, which has been widely used in multiple works (e.g., Dommain et al., 2014; Billy et al., 2018; Ma et al., 2019, 2020; Hudson et al., 2021).

Figure 4: The application of a 5 point running average has led to offsets between the timing of the actual peak and those shown in the smoothed record. A time-controlled box car filter should be applied.

Answer: Indeed, for figure 4b, the red line was calculated by the time-controlled "simple" moving average (boxcar) filter, and this point has been clarified in the revised manuscript (please see line 977). Besides, we agree that the offsets between the actual peaks and the smoothing curve. We have corrected that using a two-point moving average. Please see new figure 4.

[Figure]

Fig. 4. (a) Cd$_w$ records calculated based on the Cd/Ca of benthic foraminifera *Hoeglundina elegans* (black), *Cibicidoides pachyderma* (green), *Uvigerina peregrina* (blue), and *Globobulimina* spp. (orange) obtained from core MD77-191, (b) Cd$_w$ record from core MD77-176 reconstructed using *H. elegans* Cd/Ca, the red line is the smoothed curves using a two-point moving average. The red stars represent the modern Cd$_w$ (~0.83 nmol/kg) in the northern Indian Ocean (Boyle et al., 1995). The color shaded intervals and abbreviations are the same as in Figure 2.

Overall, the manuscript has improved a little. There are, however, issues remaining that require a further rewrite.

**Reply to reviewer #2 comments:**

Review of Ma et al. (revised version)

In my view the authors did a great job to address the concerns raised by both Reviewers. In particular they now more clearly discuss the uncertainties of their Cdw record on the potentially varying influence of primary surface productivity and intermediate water properties. Hence, I suggest acceptance of the manuscript pending very minor revisions, which basically regard typos and ambiguous wording.

In particular the abstract would benefit from more explicitly writing which factors are driving the Cdw record during which time period. The authors all state this is the abstract but it reads rather indirect. Below a recommendation for rephrasing (lines 22-29), please feel free to adopt or dismiss these suggestion.

"These results suggest that during the last deglaciation Cdw variability was primarily driven changes in intermediate water properties, indicating an enhanced ventilation of intermediate-bottom water masses during both Heinrich Stadial 1 and Younger Dryas (HS1 and YD, respectively). During the Holocene, however, surface primary productivity appeared to have influenced Cdw more than intermediate water mass properties. This is evident during the early Holocene (from 10 to 6 cal kyr BP) when benthic foraminiferal assemblages indicate that surface primary productivity was low, resulting in low intermediate water Cdw at both sites. Then, from ~

5.2 to 2.4 cal kyr BP, surface productivity increased markedly, causing a significant increase in the intermediate water Cdw in the southeastern Arabian Sea and the northeastern BoB.

Answer: We would like to thank the reviewer for this comment, providing us a more clearly way to interpreter our results in the abstract. We have corrected in the revised version, please see lines 22-36.

L. 48: "contribute to up to..:"

Answer: Corrected. Please see line 55.

L. 86: "only few works indicate the" – do you mean "investigate"?

Answer: We have corrected this sentence in the revised manuscript. Please see line 93.

L. 114: avoid using "." as multiplier

Answer: Corrected. Please see line 121.

L. 142: "linked to increased primary productivity"

Answer: Corrected. Please see line 155.

L. 148: what do you mean by "species level differences"? Inter-species offsets?

Answer: We have corrected this sentence in the revised manuscript. Please see line 161.

L. 335: "which is characterized by the well-ventilated and depleted nutrient" – something is missing here

Answer: Corrected. Please see lines 364.

L. 340: "Benthic foraminifera" - no capital

Answer: Corrected. Please see line 378.

L 357: "in the bottom water"

Answer: Corrected. Please see lines 397-398.

L. 359: "Assemblage 1" – no capital

Answer: We have corrected this sentence in the revised version. Please see lines 398-403.

L. 361: Globigerina can be abbreviated

Answer: Corrected. Please see line 399.

L. 375: "little discrepancies" – better use "small-scale" instead of little. However, I think the discrepancies are not so small, although I agree that the long-term trend is similar.

Answer: We have corrected in the revised manuscript. Please see line 420.

L. 423: "from the deep layer" – please also specify which deep layer you mean. Intermediate waters? Thermocline waters?

Answer: Corrected. Please see line 486.

L. 480-481: "another evidence for the influence of changes in water masses and/or ventilation during the HS1 and YD, as already demonstrated by" – this sentence might be rephrased as it undersells the results; especially "another evidence" and "as already" sounds like the data adds nothing new to the existing records which is not the case). I would suggest to write "… another evidence for the influence of changes in water masses and/or ventilation during the HS1 and YD, in line with…"

Answer: We have corrected in the revised manuscript. Please see line 546.

L. 498-499: "enhanced northward flow of southern sourced intermediate water mass AAIW" – there is something missing here

Answer: We have corrected this sentence in the revised manuscript. Please see line 564.

We sincerely thank the editor and reviewers for your kind considerations of our work and inspiring

suggestions for the improvement our manuscript. With your kind help, we have improved our manuscript and hope that our work meets the criteria of *Climate of the Past*.

With regards,

Sincerely,

Dr. Ruifang Ma (on behalf of all authors)